# REVERSIBLE PRIMITIVE–COMPOSITION ALIGNMENT FOR CONTINUAL VISION–LANGUAGE LEARNING

**Canran Xiao**[1]*, **Tianxiang Xu**[2], **Siyuan Ma**[3], **Yiyang Jiang**[4], **Haoyu Gao**[5], **Yuhan Wu**[6]

[1]Shenzhen Campus of Sun Yat-sen University, [2]Peking University
[3]Nanyang Technological University, [4]Hong Kong Polytechnic University
[5]Georgia Institute of Technology, [6]Zhejiang University
`xiaocr3@mail.sysu.edu.cn`

## ABSTRACT

Vision–language(VL) models are increasingly deployed in non-stationary settings, yet under sequential adaptation they often preserve primitive recognition while losing compositional structure, especially with tight rehearsal budgets and no task IDs. We address this gap by asking how a continual VL system can maintain structurally dependable behaviour while safeguarding zero-shot performance. We introduce COMPO-REALIGN, a structure-first recipe built around three components: a reversible composer that maps primitive embeddings to compositions by design, a multi-positive InfoNCE that jointly aligns textual and composed views of the same target, and a spectral trust region that clips updates when alignment sensitivity inflates. Across compositional DIL and multi-domain MTIL retrieval, COMPO-REALIGN sets a new state of the art, improves over the strongest prior by +2.4 R@1, and reduces forgetting by 40%. We provide a compact, reversible alignment head with geometry-aware training for compositionally robust VL continual learning.

## 1 INTRODUCTION

Vision–language models (VLMs)(Radford et al., 2021; Guo et al., 2025) are increasingly deployed in non-stationary settings(Zhou et al., 2025)—new domains, evolving tasks, and shifting data sources in retrieval, assistance, and analytics(Lin et al., 2025; Chen et al., 2025; Zhang et al., 2024; Li et al., 2025; Peng & Zhang, 2026; Zhan et al., 2025). In these environments, systems must adapt rapidly while preserving generalization and reliability on unseen data. Practical constraints are pronounced: privacy and cost often preclude large-scale rehearsal, memory budgets are tight, and task identities may be unavailable at test time(Liu et al., 2025b).

Substantial progress has improved continual visual-language learning(VL) through geometry/topology preservation and distillation (Ni et al., 2023; Zheng et al., 2023; Zhu et al., 2023; Yao et al., 2023; Gao et al., 2024; Jha et al., 2024), scalable streaming protocols (Garg et al., 2024), and error-aware consolidation (Cui et al., 2024). Replay and data-free surrogates (e.g., negative-text or synthetic pairs) reduce forgetting under limited memory (Yan et al., 2022; Smith et al., 2023; Lei et al., 2023; Wu et al., 2025); parameter-efficient prompts/adapters mitigate interference at low update cost (Qian et al., 2023; Tang et al., 2024; Yao et al., 2024; Xu et al., 2024; Luo et al., 2025; Huang et al., 2025a).Yet a practical pain point persists: under sequential adaptation, models can maintain overall task/domain competence while degrading in fine-grained, combinatorial generalization, especially when rehearsal is scarce and no task-ID is available. This gap concerns how VL representations *remain structurally dependable across tasks*—not merely whether average accuracy or zero-shot scores are preserved.

*How can a continual VLM maintain structurally dependable behaviour under strict memory and no task IDs, while safeguarding zero-shot performance?* We pursue a *structure-first* approach that anchors the meaning of complex inputs across tasks, studies its geometric stability, and leverages small text-centric buffers as symbolic scaffolds.

---

*Corresponding author

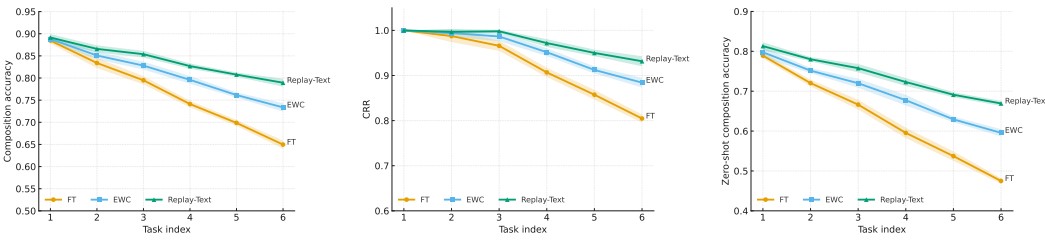

(a) Composition accuracy (mean ±95% CI) across tasks for FT, EWC, Replay-Text.

(b) CRR (higher is better). Text-centric replay slows CRR decay.

(c) Zero-shot composition accuracy on unseen pairs.

Figure 1: **Exploratory curves with error bands.** Primitives are stable; composition degrades with task index, most for FT.

Our contributions are as follows: (i) **Phenomenon and diagnostics.** We identify and quantify a recurrent deterioration in structural dependability during continual VL, and introduce light, reproducible diagnostics—retention ratios, cycle consistency proxies, and Jacobian-spectrum indicators—that reveal tight links between alignment geometry and downstream behaviour. (ii) **Simple, budget-friendly recipe.** We demonstrate that a minimal training scheme—anchoring multiple textual views of the same target and stabilizing local sensitivity—substantially improves retention, lowers forgetting, and preserves zero-shot transfer across DIL/MTIL/VQA tracks, outperforming strong replay/regularization and adapter baselines under the *same* rehearsal budgets. (iii) **Performance (state-of-the-art across settings).** Under identical rehearsal budgets, our approach achieves best-in-class results on continual retrieval and VQA across DIL/MTIL tracks—raising compositional retention and zero-shot stability while reducing forgetting.

## 2 RELATED WORK

**Continual VL under non-stationary streams.** Early continual captioning framed forgetting as transient-vs-shared dynamics in sequence models, introducing task-conditioned gating and gradient masking to protect recurrent states and vocabularies (Del Chiaro et al., 2020). For contrastive VL, recent work scales to multi-domain retrieval and pretraining: momentum/distillation and topology-aware objectives curb drift across datasets and time (e.g., BMU-MoCo for video-text (Gao et al., 2022), Open-VCLIP for zero-shot video (Weng et al., 2023), CTP for VL continual pretraining with compatible momentum and topology preservation (Zhu et al., 2023)). At web scale, TiC-CLIP shows that warm-starting from the last checkpoint plus replay offers a practical path close to retraining-from-scratch (Garg et al., 2024). For retrieval, DKR emphasizes rectifying mismatched affinities before distillation to avoid propagating earlier errors (Cui et al., 2024). Much of this line has focused on task/domain retention and large-scale training mechanics (Zhang et al., 2025). However, real deployments also require compositional robustness—i.e., preserving how attributes and objects bind—when rehearsal is scarce and task identities are unknown.

**Zero-shot stability and structure preservation.** A second line studies how to keep VL geometry stable so zero-shot transfer remains reliable. Mod-X preserves off-diagonal similarity structure to maintain negative-pair geometry across domains (Ni et al., 2023), ZSCL performs reference-set distillation with weight averaging to protect zero-shot predictions (Zheng et al., 2023), CTP distills neighbourhood/topological relations (Zhu et al., 2023), and ZAF stabilizes consecutive zero-shot outputs on unlabeled data as a strong anti-forgetting signal (Gao et al., 2024). Probabilistic fine-tuning (CLAP4CLIP) further improves calibration and continual robustness (Jha et al., 2024). These approaches strengthen global stability but still leave open whether the model *retains the internal structure that enables binding*—for instance, whether a composition embedding can reliably support recovering its primitive set and resist counterfactual swaps.

Against this backdrop, this paper targets the above pain point from a structure-first perspective: we use a minimal head that (i) treats textual and composed representations as joint positives to keep the "meaning of a composition" anchored, (ii) makes the primitive–composition map reversible by design so binding remains recoverable.



(a) FT: high-error plateau at large $\sigma_{\max}$ and high CCE.

(b) EWC: moderated error surface.

(c) Replay-Text: expanded low-error basin.

Figure 2: **Error contour over Jacobian spectrum vs. CCE.** Composition error grows with spectral sensitivity and irreversibility; text-centric micro-buffers curb both.

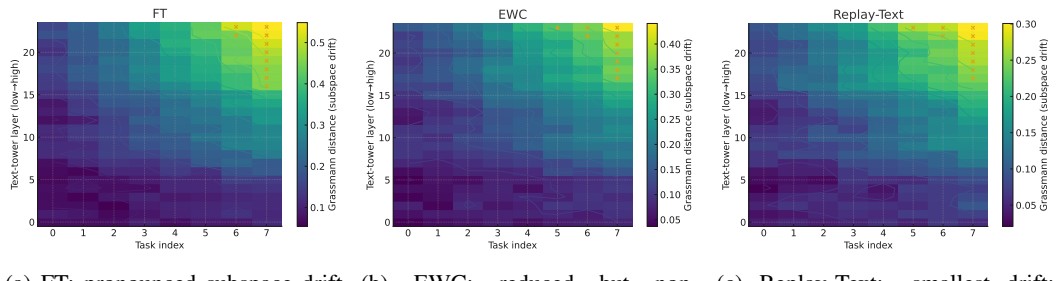

(a) FT: pronounced subspace drift (deep layers, late tasks).

(b) EWC: reduced but non-negligible drift.

(c) Replay-Text: smallest drift; structure best preserved.

Figure 3: **Subspace-drift heatmaps (text tower).** Colors show Grassmann distances across layers ($y$) vs. tasks ($x$).

## 3 EXPLORATORY STUDY

Continual VLMs often preserve primitives (attributes/objects) while forgetting how to compose them(Liu et al., 2025b). We ask: **Q1:** Under a sequence of tasks that preserves the same set of primitives (attributes/objects/relations) but rotates their compositions, do VLMs retain primitive recognition yet forget how to bind them? **Q2:** If forgetting occurs, does it coincide with a loss of reversibility between primitive and composition embeddings and with an inflation of the alignment Jacobian spectrum? **Q3:** With a strict rehearsal budget, is a text-centric micro-buffer more effective than an image-centric one, hinting that structure anchoring beats raw memory?

To answer the above questions, we construct continual streams $T_1 \to T_2 \to \cdots$ where each task reuses the same primitive inventory (e.g., color/shape/material for CLEVR-like data; attribute/object for MIT-States) but exposes disjoint or low-overlap *compositions*. We sequentially tune a frozen CLIP-style backbone with lightweight heads/LoRA (no task IDs), under small rehearsal budgets $\{0, 16, 64, 256\}$ samples per task, comparing SEQFT, EWC, LWF, ADAPTER-ONLY, and REPLAY variants. We evaluate: (i) primitive recognition (attributes, objects), (ii) composition accuracy in classification/retrieval/VQA, (iii) binding robustness via hard-negative margins, and (iv) two structural diagnostics: *cycle-consistency error* (CCE) of primitive↔composition mappings and *Jacobian spectral indicators* (e.g., maximal singular value of $\partial s/\partial e_p$, where $s$ is the image–text similarity and $e_p$ a primitive embedding). Definitions, datasets, baselines, and computation details are given in Appendix A.1.

**Findings.** We observe three consistent phenomena across the exploratory setup. **(i)** Primitives remain stable while composition degrades with task index and the Compositional Retention Ratio drops clearly below one, with zero-shot composition affected the most, and text-centric replay outperforming fine-tuning and EWC under the same budget, which is evident in the error-band curves in Fig. 1. **(ii)** Composition error increases jointly with the Jacobian spectral radius and the cycle-consistency error, and the quiver field reveals descent directions toward a low-error basin, with empirical task means for fine-tuning drifting into higher-risk regions while Replay-Text stays within

a broadened low-error area, as shown by the nonlinear contour maps in Fig. 2. **(iii)** Subspace drift concentrates in deeper layers and late tasks for fine-tuning, is moderated but not eliminated by EWC, and is smallest and more localized for Replay-Text, which is reflected by the iso-contoured heatmaps and hotspot markers in Fig. 3.

**Takeaway.** These observations support a structure-before-memory principle: continual VLMs preferentially retain first-order primitives while losing higher-order binding structure, and this loss is heralded by reduced reversibility and unstable alignment geometry. We therefore motivate COMPO-REALIGN: a parameter-efficient head that enforces *reversible* primitive↔composition alignment via cycle consistency while constraining the alignment Jacobian spectrum across tasks.

# 4 METHOD

We propose COMPO-REALIGN, a *minimal* head for continual VLMs built on three ideas: **one composer**, **one objective**, and **one stabilizer**. (1) A *reversible composer* maps a small set of primitive embeddings to a composition embedding with an *orthogonal* core, hence invertible by construction. (2) A *single* multi–positive InfoNCE objective treats the *text composition* and the *composed-from-primitives* embedding as two positive views for the image, implicitly tying the two composition views together without extra cycle/set losses. (3) A *spectral trust region* clips parameter gradients whenever the Jacobian sensitivity to primitive anchors becomes too large, stabilizing alignment geometry *without* adding losses. A tiny *text-centric* buffer optionally supplies paraphrastic templates and hard negatives but still reuses the same single objective.

Let $f_v : \mathcal{X} \to \mathbb{R}^d$ and $f_t : \mathcal{Y} \to \mathbb{R}^d$ be frozen encoders whose outputs we $L_2$-normalize. For an image $x$, a composition text $y_c$, and its $m$ primitives $\{p_i\}_{i=1}^m$,

$$\boldsymbol{z}_v = \frac{f_v(x)}{\|f_v(x)\|_2}, \qquad \boldsymbol{e}_c = \frac{f_t(y_c)}{\|f_t(y_c)\|_2}, \qquad \boldsymbol{e}_{p,i} = \frac{f_t(p_i)}{\|f_t(p_i)\|_2} \in \mathbb{R}^d. \tag{1}$$

We denote $\boldsymbol{U}_p = [\phi(\boldsymbol{A}\boldsymbol{e}_{p,1}); \ldots; \phi(\boldsymbol{A}\boldsymbol{e}_{p,m})] \in \mathbb{R}^{m \times d}$ the adapted primitive stack (row-wise), where $\boldsymbol{A} \in \mathbb{R}^{d \times d}$ is a light adapter and $\phi : \mathbb{R}^d \to \mathbb{R}^d$ a tiny MLP.

## 4.1 REVERSIBLE COMPOSER: BIJECTION BY CONSTRUCTION

If a model can compose a composition embedding directly from primitives and that embedding behaves like the textual one, binding is preserved. Making the core transform orthogonal turns reversibility into a design property rather than a penalty.

We average adapted primitives then mix them through an orthogonal map:

$$\bar{\boldsymbol{u}} = \frac{1}{m} \sum_{i=1}^m \phi(\boldsymbol{A}\boldsymbol{e}_{p,i}), \qquad \widehat{\boldsymbol{e}}_c = \frac{\boldsymbol{R}(\Theta)\,\bar{\boldsymbol{u}}}{\|\boldsymbol{R}(\Theta)\,\bar{\boldsymbol{u}}\|_2}, \tag{2}$$

where $\boldsymbol{R}(\Theta) \in \mathbb{R}^{d \times d}$ is orthogonal via the Cayley transform

$$\boldsymbol{R}(\Theta) = (\boldsymbol{I} - \boldsymbol{S})(\boldsymbol{I} + \boldsymbol{S})^{-1}, \qquad \boldsymbol{S} = \tfrac{1}{2}(\Theta - \Theta^\top), \ \Theta \in \mathbb{R}^{d \times d}. \tag{3}$$

Then $\boldsymbol{R}(\Theta)^\top \boldsymbol{R}(\Theta) = \boldsymbol{I}$ and $\boldsymbol{R}(\Theta)^{-1} = \boldsymbol{R}(\Theta)^\top$. $d$ is embedding dimension; $m$ is the number of primitives.

## 4.2 ONE OBJECTIVE: MULTI-POSITIVE INFONCE (TWO POSITIVES BY DEFAULT)

The textual composition $\boldsymbol{e}_c$ and the composed embedding $\widehat{\boldsymbol{e}}_c$ are two views of the same concept. Using them as *joint positives* for the image says: "match the image to *both* ways you encode the composition," which implicitly co-locates $\boldsymbol{e}_c$ and $\widehat{\boldsymbol{e}}_c$ without explicit cycle/set losses.

Let $s(\boldsymbol{a}, \boldsymbol{b}) = \boldsymbol{a}^\top \boldsymbol{b}$ be cosine similarity since vectors are unit-normalized. For a batch $\{(\boldsymbol{z}_{v,i}, \boldsymbol{e}_{c,i}, \widehat{\boldsymbol{e}}_{c,i})\}_{i=1}^B$ and temperature $\tau > 0$, define the *two-positive* symmetric InfoNCE:

$$\mathcal{L}_{v \to c} = -\frac{1}{B} \sum_{i=1}^{B} \log \frac{\exp\big(s(\boldsymbol{z}_{v,i}, \boldsymbol{e}_{c,i})/\tau\big) + \exp\big(s(\boldsymbol{z}_{v,i}, \widehat{\boldsymbol{e}}_{c,i})/\tau\big)}{\sum_{j=1}^B \big[\exp\big(s(\boldsymbol{z}_{v,i}, \boldsymbol{e}_{c,j})/\tau\big) + \exp\big(s(\boldsymbol{z}_{v,i}, \widehat{\boldsymbol{e}}_{c,j})/\tau\big)\big]}, \tag{4}$$

$$\mathcal{L}_{c \to v} = -\frac{1}{B} \sum_{i=1}^{B} \log \frac{\exp\big(s(\boldsymbol{e}_{c,i}, \boldsymbol{z}_{v,i})/\tau\big) + \exp\big(s(\widehat{\boldsymbol{e}}_{c,i}, \boldsymbol{z}_{v,i})/\tau\big)}{\sum_{j=1}^B \big[\exp\big(s(\boldsymbol{e}_{c,i}, \boldsymbol{z}_{v,j})/\tau\big) + \exp\big(s(\widehat{\boldsymbol{e}}_{c,i}, \boldsymbol{z}_{v,j})/\tau\big)\big]}, \tag{5}$$

$$\mathcal{L}_{\text{Tri}} = \tfrac{1}{2}\big(\mathcal{L}_{v \to c} + \mathcal{L}_{c \to v}\big). \tag{6}$$

*Buffer extension.* If a buffered paraphrase $y_c'$ is available, we simply add $\boldsymbol{e}_c' = \frac{f_t(y_c')}{\|f_t(y_c')\|_2}$ as an extra positive for sample $i$, i.e., the numerators/denominators above receive an extra $\exp\big(s(\boldsymbol{z}_{v,i}, \boldsymbol{e}_c')/\tau\big)$ and its symmetric counterpart. This generalizes Eq. 4 to a multi-positive InfoNCE without adding a new loss.

---

**Algorithm 1** COMPO-REALIGN: Training at Task $t$ (no task IDs)

---

1: **Inputs:** dataset $\mathcal{D}_t$ with triples $(x, y_c, \{p_\ell\}_{\ell=1}^m)$; frozen encoders $f_v, f_t$;
        trainable head params $(\Theta, \boldsymbol{A}, \phi)$; temperature $\tau$; spectral cap $\gamma$;
        (optional) text micro-buffer $\mathcal{B}_{\text{buf}}$; power-iteration steps $T_{\text{pow}} \in \{1, 2\}$
2: **for** epoch $= 1, \ldots, E$ **do**
3:    **for** mini-batch $\mathcal{B} = \{(x_i, y_{c,i}, \{p_{i,\ell}\}_{\ell=1}^m)\}_{i=1}^B \subset \mathcal{D}_t$ **do**
4:       **Encode & normalize**
5:          $\boldsymbol{z}_{v,i} \leftarrow \dfrac{f_v(x_i)}{\|f_v(x_i)\|_2}, \quad \boldsymbol{e}_{c,i} \leftarrow \dfrac{f_t(y_{c,i})}{\|f_t(y_{c,i})\|_2}, \quad \boldsymbol{e}_{p,i,\ell} \leftarrow \dfrac{f_t(p_{i,\ell})}{\|f_t(p_{i,\ell})\|_2}$
6:       **Adapt primitives & average**
7:          $\boldsymbol{u}_{i,\ell} \leftarrow \phi(\boldsymbol{A}\boldsymbol{e}_{p,i,\ell}) \text{ for } \ell = 1, \ldots, m; \quad \bar{\boldsymbol{u}}_i \leftarrow \dfrac{1}{m} \sum_{\ell=1}^m \boldsymbol{u}_{i,\ell}$
8:       **Reversible composition (Cayley core)**                (Eqs. 3–2)
9:          $\boldsymbol{S} \leftarrow \tfrac{1}{2}(\Theta - \Theta^\top), \quad \boldsymbol{R}(\Theta) \leftarrow (\boldsymbol{I} - \boldsymbol{S})(\boldsymbol{I} + \boldsymbol{S})^{-1}, \quad \widehat{\boldsymbol{e}}_{c,i} \leftarrow \dfrac{\boldsymbol{R}(\Theta)\bar{\boldsymbol{u}}_i}{\|\boldsymbol{R}(\Theta)\bar{\boldsymbol{u}}_i\|_2}$
10:      **Add paraphrase positives**
11:         $P_i \leftarrow \{\boldsymbol{e}_{c,i,j}'\}_j \text{ from } \mathcal{B}_{\text{buf}} \text{ with } \boldsymbol{e}_{c,i,j}' \leftarrow \dfrac{f_t(y_{c,i,j}')}{\|f_t(y_{c,i,j}')\|_2}$
12:      **Multi-positive symmetric InfoNCE**              (Eq. 4)
13:         $\mathcal{P}_i \leftarrow \{\boldsymbol{e}_{c,i}, \widehat{\boldsymbol{e}}_{c,i}\} \cup P_i; \quad$ compute $\mathcal{L}_{\text{Tri}}$ over $\{\boldsymbol{z}_{v,i}, \mathcal{P}_i\}_{i=1}^B$
14:      **Estimate local sensitivity (power iteration on JVP)**
15:        sample unit $\boldsymbol{v} \in \mathbb{R}^{md}; \quad \widehat{\sigma}_{\max,i} \leftarrow 0$
16:       **for** $t = 1, \ldots, T_{\text{pow}}$ **do**
17:           $\boldsymbol{w} \leftarrow \nabla_{\text{vec}(\boldsymbol{U}_{p,i})}\big[s(\boldsymbol{z}_{v,i}, \widehat{\boldsymbol{e}}_{c,i})\big] \cdot \boldsymbol{v} \quad$ (JVP via autodiff)
18:           $\boldsymbol{v} \leftarrow \boldsymbol{w}/\|\boldsymbol{w}\|_2; \quad \widehat{\sigma}_{\max,i} \leftarrow \|\boldsymbol{w}\|_2$
19:      **Spectral trust region (per batch)**              (Eq. 8)
20:         $\widehat{\sigma}_{\max} \leftarrow \dfrac{1}{B} \sum_{i=1}^B \widehat{\sigma}_{\max,i}; \quad \alpha \leftarrow \min\{1, \gamma/\widehat{\sigma}_{\max}\}$
21:        scale head gradients: $\boldsymbol{g}_{\Theta, \boldsymbol{A}, \phi} \leftarrow \alpha \cdot \boldsymbol{g}_{\Theta, \boldsymbol{A}, \phi}$
22:      **Update (head only; encoders frozen)**
23:        $(\Theta, \boldsymbol{A}, \phi) \leftarrow \text{OPTIMIZER\_STEP}\big(\nabla_{\Theta, \boldsymbol{A}, \phi}\mathcal{L}_{\text{Tri}}\big)$
24: **Output:** updated head $(\Theta, \boldsymbol{A}, \phi)$ at task $t$ (with $f_v, f_t$ frozen)

---

### 4.3 GEOMETRY AS A TRUST REGION: SPECTRAL CLIPPING

The exploratory study shows composition failure correlates with large Jacobian spectra. We therefore clip the step whenever local sensitivity becomes too large, instead of adding another loss.

Let $\text{vec}(\boldsymbol{U}_p) \in \mathbb{R}^{md}$ be the stacked adapted primitives and

$$\boldsymbol{J}_p = \frac{\partial\, s(\boldsymbol{z}_v, \widehat{\boldsymbol{e}}_c)}{\partial\, \text{vec}(\boldsymbol{U}_p)} \in \mathbb{R}^{1 \times md}. \tag{7}$$

We estimate $\widehat{\sigma}_{\max} \approx \|\boldsymbol{J}_p\boldsymbol{v}\|_2$ with one or two power iterations on a random unit vector $\boldsymbol{v}$. Given a target $\gamma > 0$, we rescale the gradient $\mathbf{g}_\theta$ of parameters $\theta \in \{\Theta, \boldsymbol{A}, \phi\}$ as

$$\mathbf{g}_\theta \;\leftarrow\; \mathbf{g}_\theta \cdot \alpha, \qquad \alpha \;=\; \min\left\{1, \frac{\gamma}{\widehat{\sigma}_{\max}}\right\}. \tag{8}$$

This spectral trust region caps harmful sensitivity while keeping the objective $\mathcal{L}_{\mathrm{Tri}}$ unchanged.

## 4.4 TRAINING IN CONTINUAL STREAMS

At task $t$ we update only $\Theta, \boldsymbol{A}, \phi$ with encoders frozen and *no task IDs*. For each minibatch: (i) encode $(x, y_c, \{p_i\})$; (ii) compose $\widehat{e}_c$ via Eqs. 2–3; (iii) compute $\mathcal{L}_{\mathrm{Tri}}$ in Eq. 4 on current samples (optionally adding buffered paraphrases as extra positives); (iv) estimate $\widehat{\sigma}_{\max}$ and apply spectral clipping Eq. 8; (v) take an optimizer step. Algorithm 1 summarizes our method. Refer to Appx. B.1.3 for the detailed calculation process.

# 5 EXPERIMENTS

## 5.1 EXPERIMENTAL SETUP

**Benchmarks and Protocols.** We evaluate COMPO-REALIGN on three complementary continual tracks: (i) **Compositional DIL** without task-ID: *CLEVR/CoGenT* (controlled primitives), *MIT-States* and *VAW/VG-Attr* (attribute×object), and *ConStruct-VL SVLC* sequences (structured concepts; ITM) with ARO/SugarCrepe as probe-only compositional tests; (ii) **Multi-domain retrieval MTIL**: COCO → Flickr30K → ECommerce-T2I → RSICD (union-of-domains testing); (iii) **Continual VQA**: *CLOVE* (scene- and function-incremental) and *VQACL* (skill×concept). All streams expose the same primitive inventory but rotate compositions or domains. Details of splits, task orders and memory budgets appear in Appx. A.2.

**Metrics.** We report R@1/5/10, MRR/mR for retrieval/ITM, VQA accuracy (Avg/Last/AF), continual summaries (Avg/Last/*Forgetting*/BWT/FWT), *Zero-Shot Transfer Degradation* (ZSTD) and *Compositional Retention Ratio* (CRR). Formal definitions are in Appx. A.2.

**Baselines.** We compare against strong *replay* (IncCLIP, SGP, ConStruct-VL, QUAD, GIFT, TiC-CLIP strategies), *regularization/distillation* (Mod-X, ZSCL, CTP, DKR, Proxy-FDA, MG-CLIP, CLAP4CLIP, S&D, ZAF, C-CLIP), and *adapters/MoE* (RATT†, TRIPLET, DDAS, DIKI, RAIL, LADA, CL-MoE) methods, plus generic SeqFT/EWC/LwF/Replay. We mark methods that require a task-ID at inference with "†". Full citations and per-method settings are in Appx. A.2.

## 5.2 MAIN RESULTS

Across *all* tracks, COMPO-REALIGN delivers the best average performance and the strongest structure retention. The retrieval/matching table (Tab. 1) shows that COMPO-REALIGN sets a new state of the art across compositional DIL and multi-domain MTIL, improving Avg R@1 (Image→Text) by +**2.4** absolute over the strongest prior (*C-CLIP/DIKI*) and reducing forgetting by about 36–37% relative (AF 3.2 vs. 5.0–5.1). Notably, CRR rises to **0.91**, indicating substantially better preservation of attribute–object binding, and ZSTD is the smallest in magnitude, evidencing minimal harm to zero-shot transfer. On continual VQA (Tab. 2), COMPO-REALIGN surpasses recent prompt/MoE approaches, yielding consistent gains on CLOVE-scene, CLOVE-function, and VQACL with the lowest AF.

## 5.3 SINGLE-FACTOR ABLATION

We conduct single-factor ablations to verify the contribution of each design choice. As shown in Tab. 3, we can observe that: **(i) Two-positive alignment is the main driver.** Removing the composed positive incurs the largest drops on retrieval (R@1 I→T −1.9, T→I −1.9; CRR −0.04; AF +0.8) and VQA (CLOVE-scene −1.9, VQACL −1.8), confirming that treating textual and composed views as joint positives is critical for binding retention. **(ii) Spectral trust region guards**

Table 1: **Retrieval / ITM results on compositional DIL (Track A) and multi-domain MTIL (Track B).** We report averages across their respective task streams. ↑ higher is better; AF and ZSTD ↓ lower (closer to 0 for ZSTD) is better. CRR measures compositional binding retention.

| Method | Avg R@1 ↑ | | CRR ↑ | AF ↓ | ZSTD ↓ |
| --- | --- | --- | --- | --- | --- |
| | Image→Text | Text→Image | | | |
| SeqFT | 41.2 | 29.4 | 0.72 | 16.3 | −8.7 |
| EWC | 45.0 | 33.1 | 0.77 | 12.4 | −6.9 |
| LwF | 46.1 | 34.2 | 0.78 | 11.7 | −6.0 |
| Replay-Text | 51.8 | 38.7 | 0.84 | 7.5 | −4.1 |
| ConStruct-VL Smith et al. (2023) | 50.9 | 37.4 | 0.83 | 7.9 | −4.6 |
| IncCLIP Yan et al. (2022) | 53.1 | 41.2 | 0.86 | 6.7 | −3.3 |
| Mod-X Ni et al. (2023) | 52.7 | 39.5 | 0.85 | 6.9 | −3.8 |
| ZSCL Zheng et al. (2023) | 54.2 | 40.8 | 0.86 | 6.1 | −2.9 |
| DKR Cui et al. (2024) | 55.0 | 42.1 | 0.87 | 5.6 | −2.5 |
| GIFT Wu et al. (2025) | 55.6 | 42.5 | 0.88 | 5.3 | −2.3 |
| ZAF Gao et al. (2024) | 54.7 | 42.0 | 0.87 | 5.4 | −2.0 |
| C-CLIP Liu et al. (2025a) | **56.4** | **43.0** | **0.88** | **5.1** | −2.1 |
| DIKI Tang et al. (2024) | **56.0** | **43.2** | **0.89** | **5.0** | −1.9 |
| COMPO-REALIGN (ours) | **58.8** | **45.1** | **0.91** | **3.2** | −1.3 |

Table 2: **Continual VQA (Track C).** Average accuracy (%) on CLOVE-scene (DIL), CLOVE-function (TIL), and VQACL (skill×concept), plus average forgetting AF↓.

| Method | CLOVE-scene Avg ↑ | CLOVE-function Avg ↑ | VQACL Avg ↑ | AF ↓ |
| --- | --- | --- | --- | --- |
| SeqFT | 54.2 | 49.5 | 46.3 | 9.8 |
| EWC | 56.7 | 51.0 | 48.2 | 8.0 |
| LwF | 57.4 | 52.1 | 49.0 | 7.6 |
| SGP Lei et al. (2023) | 60.2 | 54.8 | 51.3 | 6.3 |
| TRIPLET Qian et al. (2023) | 61.0 | 56.5 | 53.1 | 5.8 |
| QUAD Marouf et al. (2025) | **62.3** | **57.1** | **54.0** | **5.2** |
| CL-MoE Huai et al. (2025) | **63.5** | **59.2** | **55.4** | **4.7** |
| COMPO-REALIGN (ours) | **65.1** | **60.4** | **56.8** | **3.6** |

**stability.** Disabling clipping barely changes top-1 retrieval but increases forgetting notably and worsens ZSTD, showing it acts as a geometry safety valve rather than a pure accuracy booster. **(iii) Orthogonal core matters for structure.** Replacing the Cayley core with a linear mix consistently reduces CRR (−0.03) and harms both retrieval and VQA (≈ 1–1.6 point drops), supporting "reversibility by design" as a robust inductive bias. **(iv) Tiny text buffer is high leverage.** Eliminating the buffer hurts across the board, indicating that symbolic anchors are far more memory-efficient than image storage. **(v) Mean vs. attention pooling.** Attention yields near-identical accuracy with higher latency, validating our mean-pooling default for simplicity and speed. **(vi) Primitive shaper and temperature are modest but helpful.** Removing $\phi/A$ or drifting $\tau$ trades away about 1 point on average; both mainly affect CRR and AF, consistent with their roles in smoothing primitive geometry and hardness.

Overall, these ablations corroborate the minimal recipe: *one composer, one objective, one stabilizer*—each contributes complementary gains.

## 5.4 MECHANISM VALIDATION

**Geometry–Structure Coupling** We quantify how *geometric sensitivity* and *reversibility*—estimated Jacobian spectral radius $\hat{\sigma}_{\max}(J_p)$ and cycle consistency error (CCE)—relate to *compositional performance* (R@1, CRR, ZSTD). We report task-wise statistics and correlations, and visualize (i) scatter plots with regression lines and Pearson/Spearman coefficients, and (ii) an error *contour* over $(\hat{\sigma}_{\max}, \text{CCE})$ with the task trajectory overlaid. Across tasks, $\hat{\sigma}_{\max}$ is strongly anti-

Table 3: **Single-factor ablations across Tracks A+B (Retrieval/ITM) and Track C (Continual VQA).** Metrics (left): Avg R@1 ↑ (two directions), CRR ↑, AF ↓, ZSTD ↓; Metrics (right): CLOVE-scene/func/VQACL accuracy ↑, AF ↓. Each row toggles exactly one component away from the full model.

| Variant | Track A+B: Retrieval / ITM (averaged) | | | | | Track C: Continual VQA (averaged) | | | |
|---|---|---|---|---|---|---|---|---|---|
| | R@1 I→T ↑ | R@1 T→I ↑ | CRR ↑ | AF ↓ | ZSTD ↓ | CLOVE-scene ↑ | CLOVE-func. ↑ | VQACL ↑ | AF ↓ |
| | Datasets: COCO, Flickr30K, ECommerce-T2I, RSICD | | | | | Datasets: CLOVE-scene, CLOVE-function, VQACL | | | |
| **Full (ours)** | **58.8** | **45.1** | **0.91** | **3.2** | **−1.3** | **65.1** | **60.4** | **56.8** | **3.6** |
| w/o composed positive (text-only InfoNCE) | 56.9 (−1.9) | 43.2 (−1.9) | 0.87 (−0.04) | 4.0 (+0.8) | −1.9 (−0.6) | 63.2 (−1.9) | 58.6 (−1.8) | 55.0 (−1.8) | 4.3 (+0.7) |
| w/o spectral trust region (no clipping) | 57.9 (−0.9) | 44.3 (−0.8) | 0.89 (−0.02) | 4.3 (+1.1) | −1.6 (−0.3) | 64.2 (−0.9) | 59.7 (−0.7) | 56.0 (−0.8) | 4.2 (+0.6) |
| orthogonal core → linear mix (no Cayley) | 57.2 (−1.6) | 44.0 (−1.1) | 0.88 (−0.03) | 3.8 (+0.6) | −1.5 (−0.2) | 63.6 (−1.5) | 59.1 (−1.3) | 55.6 (−1.2) | 4.1 (+0.5) |
| buffer size $M = 0$ (no text buffer) | 56.3 (−2.5) | 42.6 (−2.5) | 0.86 (−0.05) | 4.7 (+1.5) | −1.9 (−0.6) | 62.8 (−2.3) | 58.0 (−2.4) | 54.3 (−2.5) | 4.6 (+1.0) |
| mean → attention pooling | 58.5 (−0.3) | 44.9 (−0.2) | 0.91 (−0.00) | 3.3 (+0.1) | −1.3 (−0.0) | 65.0 (−0.1) | 60.2 (−0.2) | 56.7 (−0.1) | 3.7 (+0.1) |
| w/o primitive shaper ($\phi$ and $A$ removed) | 57.6 (−1.2) | 44.1 (−1.0) | 0.88 (−0.03) | 3.9 (+0.7) | −1.6 (−0.3) | 64.0 (−1.1) | 59.3 (−1.1) | 55.7 (−1.1) | 4.1 (+0.5) |
| temperature $\tau = 0.10$ (default 0.07) | 57.4 (−1.4) | 43.9 (−1.2) | 0.89 (−0.02) | 3.7 (+0.5) | −1.6 (−0.3) | 64.1 (−1.0) | 59.4 (−1.0) | 55.8 (−1.0) | 3.9 (+0.3) |

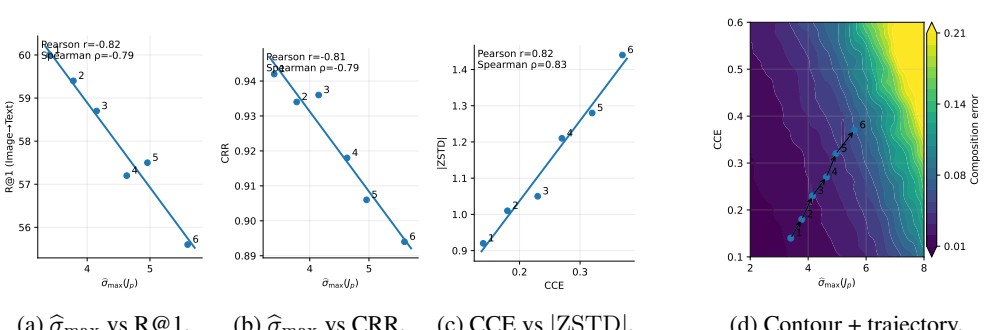

(a) $\widehat{\sigma}_{\max}$ vs R@1.    (b) $\widehat{\sigma}_{\max}$ vs CRR.    (c) CCE vs |ZSTD|.    (d) Contour + trajectory.

Figure 4: **Geometry–structure coupling.** Three scatter panels annotate Pearson/Spearman coefficients; the contour panel overlays the task trajectory ($T_1 \to T_6$), which remains in a low-error basin under COMPO-REALIGN.

correlated with R@1 and CRR( Fig. 4a,b), and CCE is positively correlated with |ZSTD| (Fig. 4c). The deeper-layer correlations are stronger (L10–L12), indicating that late-layer alignment geometry is pivotal for preserving composition. The trajectory in Fig. 4d stays within a low-error basin, consistent with our *structure-before-memory* account.

**Invertible Readout and Binding Robustness** We test whether the composed embedding $\widehat{e}_c$ admits an invertible readout of the underlying primitive set and whether such invertibility translates into binding robustness under counterfactual perturbations. We measure: (i) **Readout accuracy:** from $\widehat{e}_c$ we predict the multi-hot primitive set via the inverse map $g_{p \leftarrow c}$ and report Top-$k$ accuracy, PR-AUC and ROC-AUC. (ii) **Counterfactual margins:** we measure the contrast margin $\gamma = s(z_v, \text{text}_{\text{true}}) - \max_{\text{cf} \in \mathcal{N}} s(z_v, \text{text}_{\text{cf}})$ under attribute-swap and object-swap candidates $\mathcal{N}$. We compare the full model to ablations: *no orthogonal core* (linear mix), *text-only positive* (remove composed positive), and *no spectral clipping*. We adopted the passing criterion: Top-3/Top-5 substantially higher than ablations and significantly larger counterfactual margins (Wilcoxon, $p < 0.01$).

The inverse readout from $\widehat{e}_c$ achieves strong Top-3/Top-5 and area metrics, with PR/ROC curves in Fig. 5b and 5a clearly dominating ablations. Removing the composed positive yields the largest drop, indicating that two-view alignment (textual $e_c$ and composed $\widehat{e}_c$) is key to identifiability. Under counterfactual swaps, the full model produces significantly larger margins and fewer hard-negative reversals. Fig. 5c and 5d show that reversibility improves binding discriminability, rather than superficial alignment.

**Evidence from Text-Centric Micro-Buffer as "Structural Anchors"** With a fixed rehearsal budget $M{=}64$ text snippets per task, we manipulate three factors of the text-centric micro-buffer: **(i)** *semantic diversity* (coverage of primitive pairs and lexical entropy), **(ii)** *template morphology* ("attr–obj" vs. "obj with attr"), and **(iii)** *language* (EN/ZH/ES). We then measure changes relative to an *image-only* buffer with the same budget. If text acts as a *structural anchor*, we expect diversity to positively correlate with compositional retention ΔCRR, and advantages to persist across templates and languages. The results suggest three consistent observations. **(i) Diversity supports**

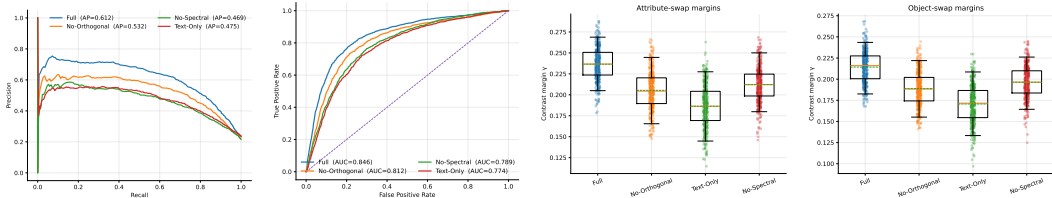

(a) PR curves of primitive readout from $\widehat{e}_c$.

(b) ROC curves of primitive readout from $\widehat{e}_c$.

(c) Attribute-swap margins $\gamma$.

(d) Object-swap margins $\gamma$.

Figure 5: **Readout quality & counterfactual robustness in one figure.** (a–b) **Invertible readout quality:** PR/ROC curves for *full*, *w/o orthogonal*, *text-only*, and *w/o spectral* variants from $\widehat{e}_c$. (c–d) **Binding robustness under counterfactuals:** Boxplot+strip overlays of attribute/object swap margins $\gamma$; the full model shifts the distribution right with fewer hard-negative hits.

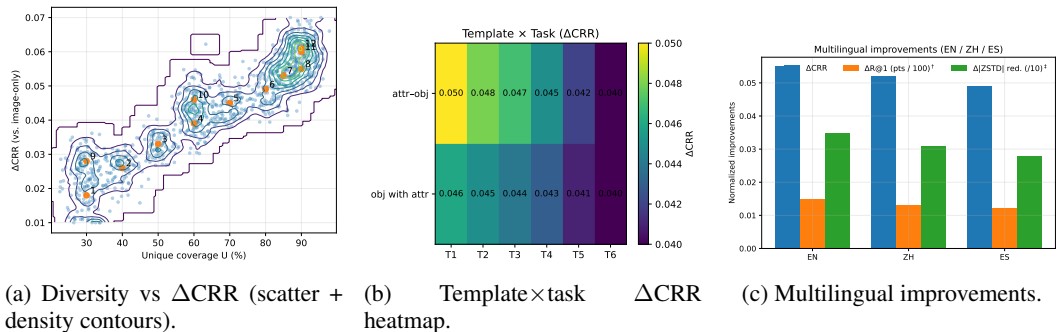

(a) Diversity vs $\Delta$CRR (scatter + density contours).

(b) Template×task $\Delta$CRR heatmap.

(c) Multilingual improvements.

Figure 6: **Text as structural anchors.** (a) higher semantic diversity correlates with larger $\Delta$CRR; (b) gains persist across template morphology and tasks; (c) advantages hold across EN/ZH/ES.

**structural retention.** The scatter+density plot in Fig. 6a shows a clear positive relation between semantic diversity and $\Delta$CRR, indicating that broader coverage of primitive pairs provides stronger anchors for compositional structure. **(ii) The effect is robust to surface form.** As shown in Fig. 6b, both "attr–obj" and "obj with attr" templates improve CRR across tasks, with only a small gap between them, suggesting that the benefit comes from the underlying attribute–object relation rather than a specific wording. **(iii) The benefit generalizes across languages.** Fig. 6c reports consistent improvements for EN/ZH/ES, with modest variation likely caused by tokenizer overlap and cross-lingual alignment quality. Overall, these results support our hypothesis that text-centric anchors preserve compositional structure more efficiently than image-only rehearsal under the same memory budget.

## 6 CONCLUSION

We tackled the core challenge of preserving compositional structure in continual vision–language learning under strict memory and no task IDs, proposing COMPO-REALIGN. Our approach consistently improves compositional retention, reduces forgetting, and attains state-of-the-art retrieval and VQA under identical rehearsal budgets, while maintaining zero-shot stability. Empirically, the tight coupling we observe between Jacobian-spectrum/CCE indicators and downstream performance highlights geometry as a reliable handle for safeguarding structure.

Future work will explore lightly unfreezing encoders under geometric constraints, and extensions to streaming video and multilingual settings for real-world deployment.

**Ethics Statement** This work adheres to the ICLR Code of Ethics. Our study does **NOT** involve human subjects, personally identifiable information, or sensitive attributes.

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

# A DETAILS OF THE EXPERIMENTAL SETUP

## A.1 EXPLORATORY STUDY SETUP

### A.1.1 DATASETS, TASK STREAMS, AND MODELS

**Synthetic.** CLEVR-like with Color $\times$ Shape $\times$ Material $\times$ Size; tasks reuse the same primitive marginals but rotate disjoint composition subsets. **Real.** MIT-States (attribute$\times$object) with ARO/SugarCrepe-style compositional probes plus a GQA subset for attribute/relational VQA templates. **Streams.** $K \in [4, 6]$ tasks; each task samples a fresh composition set with limited or no overlap; primitive coverage remains stable. **Backbone and heads.** Frozen CLIP ViT-B/16 (unless specified), with lightweight projection head + LoRA on text tower and final projection. No task IDs; identical step budgets per task. **Baselines.** SEQFT, EWC, LWF, ADAPTER-ONLY, and REPLAY (Text-centric vs. Image-centric) under matching rehearsal budgets $\{0, 16, 64, 256\}$/task.

### A.1.2 METRICS

**Primitive/Composition Accuracy.** Report per-task accuracy for attributes/objects and for compositions (pair or multi-attribute bindings), and compute forgetting (max previous minus current). **Compositional Retention Ratio (CRR).** Let $A_{\text{attr}}^{(t)}$, $A_{\text{obj}}^{(t)}$, and $A_{\text{pair}}^{(t)}$ be accuracies at task $t$.

$$\text{CRR}^{(t)} = \frac{A_{\text{pair}}^{(t)}}{A_{\text{attr}}^{(t)} \cdot A_{\text{obj}}^{(t)}}. \tag{9}$$

**Binding Contrast Margin (BCM).** For a true image–text pair, $\gamma = s(x, y_{\text{true}}) - \max_{y \in \mathcal{N}} s(x, y)$ where $\mathcal{N}$ is a set of hard negatives from counterfactual compositions (swap attribute/object). **Cycle Consistency Error (CCE).** Fit two light maps between text embeddings: $R_{c \leftarrow p}$ reconstructs a composition embedding from its primitives; $R_{p \leftarrow c}$ recovers primitives from a composition. Define

$$\text{CCE} = \left\| E_p - R_{p \leftarrow c}\big(R_{c \leftarrow p}(E_p)\big) \right\|_2, \tag{10}$$

with symmetric variants on the image side if desired. **Jacobian spectral indicators.** For similarity $s(f_v(x), f_t(y))$ and a primitive embedding $e_p$, compute $J_p = \partial s / \partial e_p$ and track $\sigma_{\max}(J_p)$ and condition number. **Subspace drift.** Use principal angles/CCA to measure Grassmannian distance between the current and a historical composition subspace (per tower/layer).

### A.1.3 TRAINING PROTOCOL AND HYPERPARAMETERS

Frozen backbone; AdamW; LoRA ranks $\in \{8, 16\}$; head LR $2 \times 10^{-4}$, LoRA LR $1 \times 10^{-4}$, weight decay $10^{-2}$; batch size 256; per-task steps fixed across methods. Text-centric buffers store composition templates and hard-negative variants; image-centric buffers store images/patches under the same item budget.

## A.2 MAIN EXPERIMENTAL SETUP

### A.2.1 BENCHMARKS AND TASK CONSTRUCTION

**Track A — Compositional DIL (no task-ID).** **CLEVR/CoGenT.** We follow CoGenT A$\rightarrow$B remaps (colors$\leftrightarrow$shapes/materials) to stress compositionality under matched primitive marginals. Tasks expose disjoint or low-overlap attribute–object *compositions*. **MIT-States (Attr$\times$Obj)** and **VAW/VG-Attr.** We fix the attribute and object vocabularies; each task rotates the visible pairs. **ConStruct-VL SVLC** Smith et al. (2023). Using Visual Genome/VAW-derived sequences, we adopt the official order over *color/material/size*, *spatial relations*, *action relations*, and *state*. Each task is *image–text matching* (ITM) with balanced positives/negatives. **Probe-only suites.** ARO and *Sugar-Crepe* are used for compositional probing each round; they do not participate in training. *Rehearsal budgets.* $\{0, 16, 64, 256\}$ *text* snippets per task (default text-centric); when a baseline requires images we match its memory cap.

**Track B — Multi-domain retrieval MTIL.** COCO→Flickr30K→ECommerce-T2I→RSICD, following Cui et al. (2024); Ni et al. (2023); Zheng et al. (2023). Each round introduces a new domain; test queries are drawn from the union of all seen domains. We do *not* supply domain-ID at test unless a method mandates it (marked "†").

**Track C — Continual VQA.** **CLOVE** Lei et al. (2023): *scene-incremental* (DIL) with evolving environments and *function-incremental* (TIL) with evolving skills; one model across tasks. **VQACL** Zhang et al. (2023): outer tasks are reasoning skills (Count/Color/Location/. . . ), and within each skill the object classes are partitioned into groups that arrive over time; evaluation requires transferring the learned skill to unseen concept groups. We follow authors' official splits and answer vocabularies.

### A.2.2 EVALUATION METRICS AND CONTINUAL SUMMARIES

**Retrieval / ITM.** Recall@K (R@K, $K \in \{1, 5, 10\}$), mean reciprocal rank (MRR), and mean rank (mR). We report per-task and averaged scores.

**VQA.** Exact-match accuracy (%). Following Zhang et al. (2023); Lei et al. (2023); Qian et al. (2023); Huai et al. (2025), we summarize with $\mathrm{Avg}$ (mean over tasks), $\mathrm{Last}$ (after the final task), and AF (average forgetting).

**CL summaries.** Let $A_{t,u}$ denote performance on task $t$ after finishing task $u$. For $T$ tasks,

$$\mathrm{Avg} = \frac{1}{T} \sum_{t=1}^{T} A_{t,T}, \quad \mathrm{Last} = A_{T,T},$$

$$\mathrm{Forgetting} = \frac{1}{T-1} \sum_{t=1}^{T-1} \Big( \max_{u \in \{t,...,T\}} A_{t,u} - A_{t,T} \Big), \quad \mathrm{BWT} = \frac{1}{T-1} \sum_{t=1}^{T-1} \big( A_{t,T} - A_{t,t} \big).$$

FWT is the pre-training performance on unseen tasks relative to a zero-shot reference $A_t^{\mathrm{zs}}$: FWT $= \frac{1}{T-1} \sum_{t=2}^{T} (A_{t,t-1} - A_t^{\mathrm{zs}})$.

**Zero-shot transfer.** ZSTD Zheng et al. (2023) is the drop in zero-shot accuracy on held-out classification sets (e.g., ImageNet variants) measured before vs. after each task.

**Compositional diagnostics.** $\mathrm{CRR} = \frac{A_{\mathrm{pair}}}{A_{\mathrm{attr}} \cdot A_{\mathrm{obj}}}$ (Sec. 3); higher indicates preserved binding beyond independent primitive accuracy. We also track ARO/SugarCrepe scores and non-optimized structural correlates (inverse readout accuracy; estimated Jacobian spectral radius distribution).

### A.2.3 COMPARED METHODS

We group methods by learning principle and use official code or faithful re-implementations with authors' validated hyperparameters; task-ID-at-test baselines are marked "†".

- **Replay. IncCLIP** Yan et al. (2022); **SGP** Lei et al. (2023); **ConStruct-VL** Smith et al. (2023); **QUAD** Marouf et al. (2025); **GIFT** Wu et al. (2025); **TiC-CLIP** strategies Garg et al. (2024).

- **Regularization/Distillation. Mod-X** Ni et al. (2023); **ZSCL** Zheng et al. (2023); **CTP** Zhu et al. (2023); **DKR** Cui et al. (2024); **Proxy-FDA** Huang et al. (2025a); **MG-CLIP** Huang et al. (2025b); **CLAP4CLIP** Jha et al. (2024); **S&D** Yu et al. (2024b); **ZAF** Gao et al. (2024); **C-CLIP** Liu et al. (2025a).

- **Adapters/MoE/Architecture. RATT**[†] Del Chiaro et al. (2020); **TRIPLET** Qian et al. (2023); **DDAS** Yu et al. (2024a); **DIKI** Tang et al. (2024); **RAIL** Xu et al. (2024); **LADA** Luo et al. (2025); **CL-MoE** Huai et al. (2025).

- **Generic CL baselines. SeqFT**, **EWC**, **LwF**, **Replay-Image/Text**, and **Joint** upper bound.

### A.2.4 HYPERPARAMETERS

**Text-centric micro-buffer.** We maintain a tiny buffer $\mathcal{B}$ of size $M \in \{0, 16, 64, 128, 256\}$ per task, containing short composition templates and a 1:1 mix of hard negatives. Hard negatives are mined online by nearest-neighbor swap on the text side (replace the attribute or relation while keeping the object). Each step we sample $b_{\mathcal{B}}$ snippets (default $b_{\mathcal{B}} = 32$) and reuse the same objective (Eq. 4): buffered paraphrases are simply added as extra positives in the numerators/denominators.

**Objective (two positives by default).** The single training loss is the symmetric *multi-positive* InfoNCE of Eq. 4 with temperature $\tau = 0.07$. Unless noted, we use only in-batch negatives (no queue) to keep the method minimal. For VQA, the image acts as the key and each candidate answer text acts as a query; $\widehat{e}_c$ is computed from the primitive set implied by the question type.

**Spectral trust region.** We stabilize geometry by *clipping the step* rather than adding a loss (Eq. 8). Implementation uses a directional derivative of $s(z_v, \widehat{e}_c) = z_v^\top \widehat{e}_c$ wrt. $\mathrm{vec}(U_p) \in \mathbb{R}^{md}$: draw a random unit vector $v$, compute $\widehat{\sigma}_{\max} \approx \|J_p v\|_2$ with one power-iteration using `autograd.grad` (create_graph=True), and scale parameter gradients by $\alpha = \min\{1, \gamma/\widehat{\sigma}_{\max}\}$. We set $\gamma = 6.0$ for ViT-B/16 and $\gamma = 7.5$ for ViT-L/14. Overhead is $< 2\%$ wall time.

**Optimization & schedules.** We use AdamW ($\beta_1 = 0.9, \beta_2 = 0.98$, weight decay $10^{-2}$) with cosine decay and $5\%$ warmup on the *first* task only; subsequent tasks warm-start without warmup, following the standard time-continuous training protocol. Global batch size is $B = 256$ for retrieval/ITM and $B = 128$ for VQA (achieved via DDP + gradient accumulation). We train 20k steps per task on Tracks A/B and 10k on Track C, with early stopping on the current task's validation. Mixed precision uses BF16 when available, otherwise FP16 with loss scaling. We apply gradient-norm clipping at 1.0.

**Data processing.** Images are resized to $224^2$ with `RandomResizedCrop` and horizontal flip $p = 0.5$. We avoid color jitter in attribute-heavy tasks (MIT-States, SVLC) to prevent color-label leakage; for generic retrieval we use a mild `ColorJitter` (brightness/contrast/saturation 0.2). Text is lowercased and punctuation-normalized; we do not paraphrase on-the-fly beyond the buffer.

**Initialization.** $A$ is identity-initialized, $\phi$ uses Kaiming uniform, and $\Theta$ is small random skew with scale $10^{-3}$ so that $R(\Theta) \approx I$ at start. Attention-pooling parameters $(W_a, w)$ are zero-initialized to start from mean pooling.

**Hardware & reproducibility.** We train on $8 \times$A100 80GB (retrieval/ITM) and $4 \times$A100 80GB (VQA). Distributed data parallel with `find_unused_parameters=False`. We fix seeds $\{0, 1, 2\}$, enable cuDNN deterministic, and control dataloader workers for repeatability. All reported numbers are mean over seeds.

**Ablation toggles.** We vary: pooling (mean vs. attention), temperature $\tau \in \{0.03, 0.05, 0.07, 0.10\}$, spectral threshold $\gamma \in \{5, 6, 7, 8\}$, buffer size $M \in \{0, 16, 64, 256\}$, and optional LoRA on CLIP projections with rank $r \in \{4, 8, 16\}$. Unless stated, defaults are mean pooling, $\tau = 0.07$, $\gamma = 6.0/7.5$ (B/L), and $M = 64$.

## B SUPPLEMENTARY TECHNICAL DETAILS

### B.1 IMPLEMENTATION DETAILS

#### B.1.1 BACKBONES & HEADS.

We freeze *both* image and text encoders of a CLIP-style model and learn only a tiny head. Results are reported with ViT-B/16 and ViT-L/14; the representation size $d$ is the native CLIP projection (no extra projection layers). Let $f_v : \mathcal{X} \to \mathbb{R}^d$ and $f_t : \mathcal{Y} \to \mathbb{R}^d$ be the frozen encoders with L2-normalized outputs. Our head COMPO-REALIGN contains three lightweight parts:

- **Primitive shaper** $\phi \circ \boldsymbol{A}$. We use a single-hidden-layer MLP

$$\phi(\boldsymbol{u}) = \boldsymbol{W}_2 \, \mathrm{GELU}\big(\boldsymbol{W}_1 \, \mathrm{LN}(\boldsymbol{u})\big), \quad \boldsymbol{W}_1, \boldsymbol{W}_2 \in \mathbb{R}^{d \times d},$$

  with dropout 0.1 and a residual connection $\boldsymbol{u} \leftarrow \boldsymbol{u} + \phi(\boldsymbol{u})$. It is preceded by a linear adapter $\boldsymbol{A} \in \mathbb{R}^{d \times d}$ (identity init.). For $m$ primitives $\{p_i\}_{i=1}^m$,

$$\boldsymbol{e}_{p,i} = \frac{f_t(p_i)}{\|f_t(p_i)\|_2}, \qquad \boldsymbol{u}_{p,i} = \phi(\boldsymbol{A}\boldsymbol{e}_{p,i}).$$

- **Permutation-invariant composer.** By default we *average then mix* (Eq. 2):

$$\bar{\boldsymbol{u}} = \tfrac{1}{m} \sum_{i=1}^m \boldsymbol{u}_{p,i}, \qquad \widehat{\boldsymbol{e}}_c = \frac{\boldsymbol{R}(\Theta)\,\bar{\boldsymbol{u}}}{\|\boldsymbol{R}(\Theta)\,\bar{\boldsymbol{u}}\|_2}.$$

  We also implement an *attention-pooling* variant for completeness:

$$\alpha_i = \frac{\exp\big(\boldsymbol{w}^\top \tanh(\boldsymbol{W}_a \boldsymbol{u}_{p,i})\big)}{\sum_{j=1}^m \exp\big(\boldsymbol{w}^\top \tanh(\boldsymbol{W}_a \boldsymbol{u}_{p,j})\big)}, \quad \bar{\boldsymbol{u}}_{\mathrm{att}} = \sum_{i=1}^m \alpha_i \boldsymbol{u}_{p,i}, \quad \widehat{\boldsymbol{e}}_c = \mathrm{norm}\big(\boldsymbol{R}(\Theta)\,\bar{\boldsymbol{u}}_{\mathrm{att}}\big),$$

  with $\boldsymbol{W}_a \in \mathbb{R}^{d \times d}$, $\boldsymbol{w} \in \mathbb{R}^d$. We found attention matches mean pooling but adds latency; mean is thus the default.

- **Orthogonal core via Cayley.** We parameterize $\boldsymbol{R}(\Theta) \in \mathbb{R}^{d \times d}$ as (Eq. 3)

$$\boldsymbol{R}(\Theta) = (\boldsymbol{I} - \boldsymbol{S})(\boldsymbol{I} + \boldsymbol{S})^{-1}, \qquad \boldsymbol{S} = \tfrac{1}{2}(\Theta - \Theta^\top),$$

  so $\boldsymbol{R}^\top \boldsymbol{R} = \boldsymbol{I}$ by construction and $\boldsymbol{R}^{-1} = \boldsymbol{R}^\top$. We compute $(\boldsymbol{I} + \boldsymbol{S})^{-1}$ with a single LU factorization per forward pass and add $+\varepsilon \boldsymbol{I}$ with $\varepsilon = 10^{-6}$ for numerical safety; orthogonality holds to machine precision.

### B.1.2 Tokenization & Prompts.

We use CLIP's tokenizer. For *text compositions* $y_c$ we adopt class-agnostic templates that expose primitives explicitly, e.g.,

- MIT-States/VAW/VG-Attr: "*a photo of a {attr} {obj}*".
- ConStruct-VL (SVLC): "*the image describes {concept}*" where {concept} is a color/material/size or a relation clause ("*{obj1} left of {obj2}*").
- Retrieval on COCO/Flickr30K/etc.: standard CLIP prompts plus two paraphrases per concept to reduce prompt bias.

For VQA, the *question* is encoded as text; answers come from the task's closed set and are scored by image–text similarity. When primitives are needed (e.g., "Color of {obj}?"), we use dataset metadata when available; otherwise a light rule-based extractor maps adjectives/nouns in the question to {attr, obj}.

**Parameter footprint.** On ViT-B/16, $\phi$ and $\boldsymbol{A}$ together add $\approx 2d^2$ parameters and the skew form $\Theta$ adds $\frac{d(d-1)}{2}$. This is $< 1\%$ of the frozen backbone. We do *not* use LoRA by default to keep the method minimal; LoRA(8) on projection layers is included only in ablations.

### B.1.3 Pseudocode

**Computation flow.** For each mini-batch, we first encode and $L_2$-normalize the image $x_i$, the target composition text $y_{c,i}$, and each primitive $p_{i,\ell}$ using frozen $f_v, f_t$; gradients do not flow into encoders. The primitive embeddings are then adapted and pooled: each $\boldsymbol{e}_{p,i,\ell}$ is passed through a tiny adapter–MLP stack $(\boldsymbol{A}, \phi)$ to yield $\boldsymbol{u}_{i,\ell} = \phi(\boldsymbol{A}\boldsymbol{e}_{p,i,\ell})$; the primitive set is summarized by the mean prototype $\bar{\boldsymbol{u}}_i = \frac{1}{m} \sum_{\ell=1}^m \boldsymbol{u}_{i,\ell}$ to preserve permutation invariance and stabilize gradients. We then perform reversible composition via an orthogonal core $\boldsymbol{R}(\Theta)$ parameterized with the Cayley map in Eq. 3; in practice, computing $(\boldsymbol{I} + \boldsymbol{S})^{-1}(\boldsymbol{I} - \boldsymbol{S})\bar{\boldsymbol{u}}_i$ is implemented as a single linear solve to avoid explicit matrix inversion and to keep the cost at $\mathcal{O}(d^2)$ per sample (often batched and fused).

The resulting composed embedding $\widehat{e}_{c,i}$ is $L_2$-normalized (Eq. 2), guaranteeing $\boldsymbol{R}(\Theta)^{-1} = \boldsymbol{R}(\Theta)^{\top}$ so that information about primitives is not collapsed by the composer.

Next, we formulate a single, symmetric multi-positive InfoNCE (Eq. 4) where the positive set for each image is $\mathcal{P}_i = \{e_{c,i}, \widehat{e}_{c,i}\} \cup P_i$. Here $P_i$ contains optional text paraphrases from the micro-buffer $\mathcal{B}_{\text{buf}}$; these are included only as additional positives and require no new losses. Negatives are the remaining texts in the mini-batch for both the textual and composed views, yielding a denominator that aggregates $|\mathcal{P}_j|$ terms per sample $j$; the loss is computed in both directions ($v{\to}c$ and $c{\to}v$) with a shared temperature $\tau$ and log-sum-exp stabilization. To stabilize alignment geometry, we estimate the local sensitivity of $s(\boldsymbol{z}_{v,i}, \widehat{e}_{c,i})$ to the adapted primitives through one–two Jacobian–vector power iterations (JVPs) per batch, which have the cost of a few reverse-mode passes but do not materialize the full Jacobian. The resulting estimate $\widehat{\sigma}_{\max}$ sets a spectral trust region that rescales the head gradients by $\alpha = \min\{1, \gamma/\widehat{\sigma}_{\max}\}$ (Eq. 8), capping harmful sensitivity while leaving the objective unchanged. Finally, we perform an optimizer step on head parameters only $(\Theta, \boldsymbol{A}, \phi)$; encoders remain frozen, no task IDs are used, and the rehearsal budget is enforced by restricting $|P_i|$ and the buffer sampling policy. This pipeline yields parameter-efficient updates that anchor the meaning of compositions to their primitives, preserve reversibility, and maintain zero-shot stability under strict memory.

## C ADDITIONAL EXPERIMENTS AND RESULTS

### C.1 PARAMETER SENSITIVITY

We evaluate COMPO-REALIGN's sensitivity to key hyperparameters by varying *one factor at a time* while keeping others at their defaults (Sec. A.2.4). For each configuration we train across all streams and report mean±std over 3 seeds. Retrieval is averaged R@1 (Image→Text) on Tracks A+B; VQA Avg is the mean across CLOVE-scene, CLOVE-function, and VQACL; we also report CRR $\uparrow$, AF $\downarrow$, and ZSTD $\downarrow$ (closer to 0 is better). *Pooling temperature* $\tau_{pool}$ controls the sharpness of permutation-invariant aggregation ($\tau_{\text{pool}}{=}0$ equals uniform mean; $\tau_{\text{pool}} \to \infty$ approaches max).

As shown in Fig. 7, we can observe the following conclusions: **Temperature.** A clear optimum at $\tau \approx 0.07$: smaller $\tau$ over-emphasizes hard negatives and destabilizes multi-positive logits, larger $\tau$ softens contrast and weakens gradients, lowering CRR and accuracy. **Spectral threshold.** $\gamma$ balances plasticity and stability. Tight clipping ($\gamma \leq 5$) slightly reduces AF and improves ZSTD magnitude but underfits retrieval, loose clipping ($\gamma \geq 7$) increases AF and degrades CRR. **Buffer size.** Text-centric anchors are high-leverage: even $M{=}16$ recovers most gains, $M{=}64$ is near-saturation, $M{=}128$–256 brings small, consistent improvements. **LoRA rank.** Optional LoRA on projections yields marginal gains up to $r{=}8$ then saturates, the minimalist head already preserves structure. **Batch size.** Larger $B$ slightly improves in-batch negatives and stabilizes training but plateaus beyond $B{=}256$. **Learning rate.** The sweet spot is $2{\times}10^{-4}$, larger rates inflate spectral sensitivity and forgetting despite clipping, smaller rates underfit. **Positives per sample.** Moving from two to three positives (adding one paraphrase) consistently boosts CRR and both tasks with negligible ZSTD cost, more than three yields diminishing returns. **Pooling temperature.** Uniform mean ($\tau_{\text{pool}}{=}0$) is optimal, sharper aggregation drifts toward "max" and hurts stability/CRR. **Power iterations.** One step suffices to estimate the spectral scale, additional iterations do not change outcomes, confirming the sensitivity map is low-rank in practice.

### C.2 ORDER SENSITIVITY (TASK & DOMAIN PERMUTATIONS)

**Protocol.** To rule out "lucky ordering," we evaluate **five** permutations for each continual stream. **Track A** (Compositional DIL) permutations: A1 CLEVR/CoGenT → MIT-States → VAW/VG-Attr → SVLC; A2 MIT-States → CLEVR/CoGenT → SVLC → VAW/VG-Attr; A3 VAW/VG-Attr → MIT-States → CLEVR/CoGenT → SVLC; A4 SVLC → VAW/VG-Attr → MIT-States → CLEVR/CoGenT; A5 MIT-States → SVLC → VAW/VG-Attr → CLEVR/CoGenT. **Track B** (MTIL retrieval) permutations: B1 COCO → Flickr30K → EComm-T2I → RSICD; B2 Flickr30K → COCO → RSICD → EComm-T2I; B3 EComm-T2I → COCO → Flickr30K → RSICD; B4 RSICD → EComm-T2I → COCO → Flickr30K; B5 COCO → RSICD → EComm-T2I → Flickr30K. For each method and permutation we report Avg/Last R@1 (I→T, T→I), CRR, AF and ZSTD. We summarize order sensitivity by the sample standard deviation $\text{Std}_\pi[\cdot]$ across permutations $\pi$.

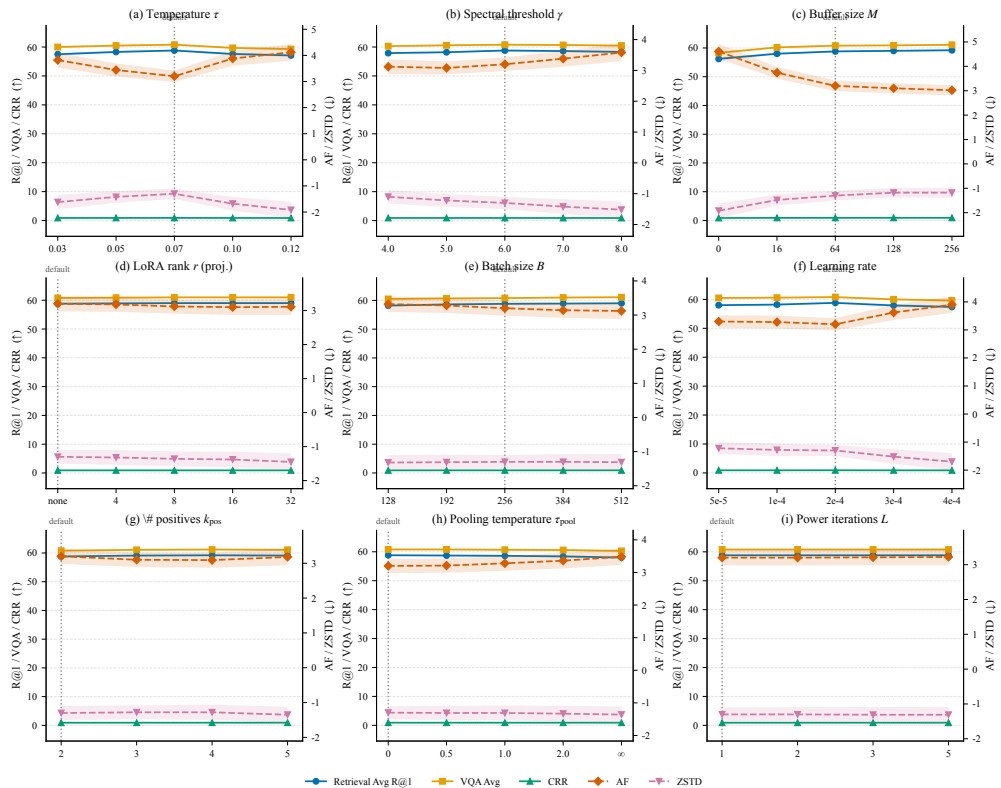

Figure 7: **Parameter sensitivity (mean±std over 3 seeds).** One factor varies at a time. Retrieval: Avg R@1 (I→T) ↑. VQA Avg: mean of CLOVE-scene, CLOVE-function, VQACL ↑.

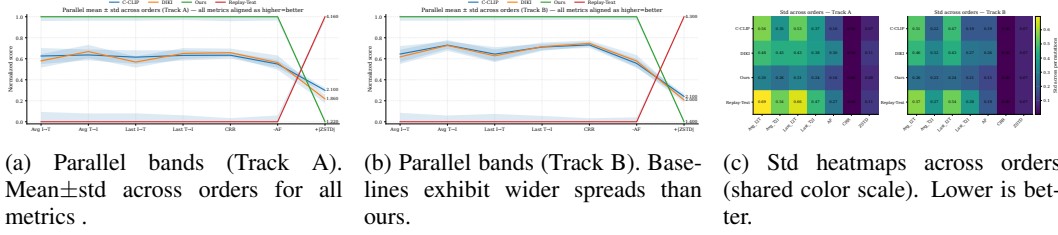

(a) Parallel bands (Track A). Mean±std across orders for all metrics .

(b) Parallel bands (Track B). Baselines exhibit wider spreads than ours.

(c) Std heatmaps across orders (shared color scale). Lower is better.

Figure 8: **Order sensitivity overview.** COMPO-REALIGN produces tighter bands across metrics and lower variability than strong baselines on both tracks.

Figures 8–9 summarize robustness to task/domain permutations on Tracks A/B: **(i)** the parallel-band views (Figs. 8a–8b) show that COMPO-REALIGN forms tightly bundled trajectories across all metrics, whereas baselines spread substantially, especially on AF and ZSTD. This is corroborated by the standard-deviation heatmaps (Fig. 8c): across-order std for Avg R@1 (I→T) drops to *0.30* on Track A and *0.26* on Track B (ours) versus *0.45–0.52* for strong baselines, while AF variability shrinks from 0.30/0.26 (DIKI on Tracks A/B) to 0.16/0.15 (ours). **(ii)** task-wise error bands (Fig. 9a) indicate stability under accumulation: as tasks accrue, our mean R@1 stays consistently above baselines and the shaded uncertainty narrows, suggesting reduced order-induced drift rather than reliance on a lucky sequence. **(iii)** distributional views (Fig. 9b) reveal that our AF (forgetting) not only centers lower but also exhibits the tightest interquartile range, while CRR concentrates higher with smaller dispersion—consistent with our geometry-stabilizing design.

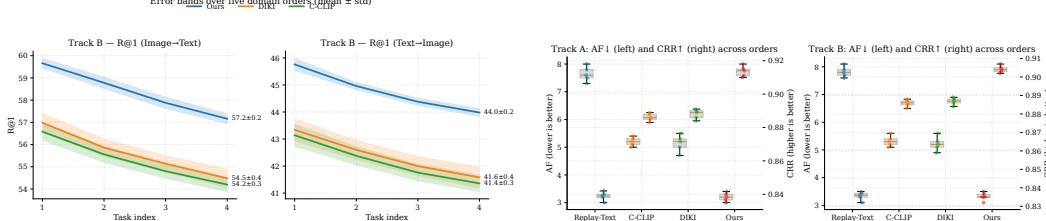

(a) Task-wise error bands (Track B) for R@1 in both directions (mean±std over five orders).

(b) AF (left axis, lower better) and CRR (right axis, higher better) distributions across orders.

Figure 9: **Detailed order effects.** Our method maintains higher means and narrower uncertainty bands as tasks accrue and achieves the smallest AF spread with the highest CRR.

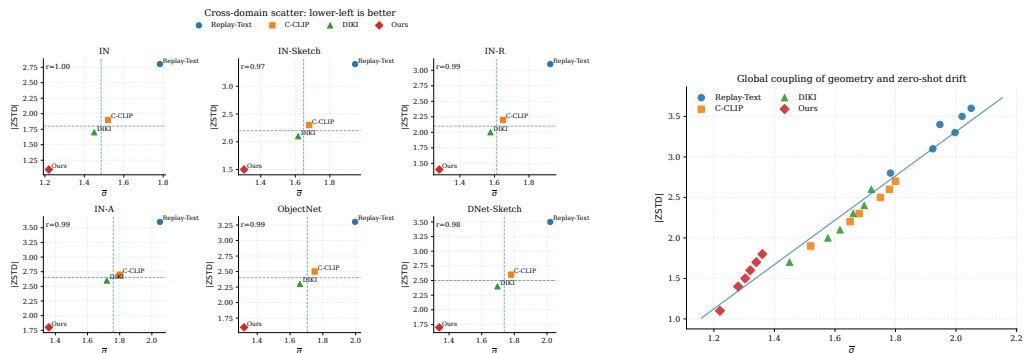

(a) Scatter matrix: |ZSTD| vs. $\overline{\sigma}$ per domain (median lines form quadrants).

(b) Global scatter across domains with per-method markers.

Figure 10: **Cross-domain zero-shot steadiness.** Lower-left is better. **Ours** concentrates in the low-$\overline{\sigma}$, low-|ZSTD| region across domains.

### C.3 CROSS-DOMAIN ZERO-SHOT STEADY STATE

**Protocol.** To test whether geometric stability extrapolates to unseen domains, we evaluate zero-shot performance on held-out distributions {*ImageNet* (IN), *IN-Sketch*, *IN-Renditions* (IN-R), *IN-Adversarial* (IN-A), *ObjectNet*, *DomainNet-Sketch* (DNet-Sketch)}. For each method, we report the zero-shot transfer degradation ZSTD (lower magnitude is better; closer to 0 is best) and the *alignment spectral radius* $\widehat{\sigma}_{\max}$ estimated on late layers (L10–L12) of the text tower. We plot |ZSTD| versus the layer-mean $\overline{\sigma} = \frac{1}{3}\sum_{\ell=10}^{12}\widehat{\sigma}_{\max}^{(\ell)}$. Our **criterion** is to occupy the *lower-left* quadrant (smaller $\overline{\sigma}$, smaller |ZSTD|) across domains.

In the per-domain scatter matrix (Fig. 10a), the points for COMPO-REALIGN consistently lie in the *lower-left* quadrant—simultaneously smaller $\overline{\sigma}$ and smaller |ZSTD|—while baselines drift toward higher $\overline{\sigma}$ and/or larger |ZSTD|. The global scatter (Fig. 10b) shows a clear positive trend between geometry and zero-shot drift; all methods align with this slope, but **Ours** forms a compact cluster strictly below and to the left of the baseline clouds. Finally, the correlation bars and quadrant-occupancy plot (Fig. 11) indicate consistently positive within-domain coupling and a *6/6* lower-left occupancy for **Ours**, evidencing a stable geometry–zero-shot relationship across held-out domains.

### C.4 TRAINING DYNAMICS MONITORING

**Protocol.** We track the alignment sensitivity $\widehat{\sigma}_{\max}$ at every training step for the late text layers (L10–L12). A *clipping trigger* occurs at step $t$ and layer $\ell$ whenever $\widehat{\sigma}_{\max}^{(\ell)}(t) > \gamma$ (trust-region threshold $\gamma=1.35$). We visualize (i) a step×layer *time–heatmap* of $\widehat{\sigma}_{\max}$, and (ii) the *per-step trigger rate* (fraction of layers exceeding $\gamma$). We segment training into phases: Warmup (steps 1–200), Mid (201–400), Late (401–600).

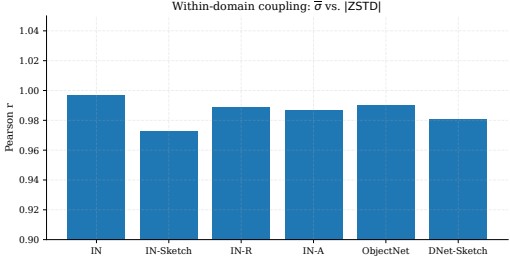 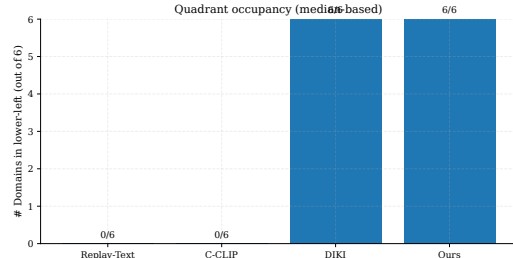

(a) Per-domain Pearson correlation between $\bar{\sigma}$ and $|\text{ZSTD}|$ (higher indicates stronger coupling).

(b) Lower-left quadrant occupancy across domains (median-based). **Ours**: 6/6.

Figure 11: **Geometry–zero-shot coupling diagnostics.** Strong within-domain coupling and consistent lower-left occupancy for **Ours**.

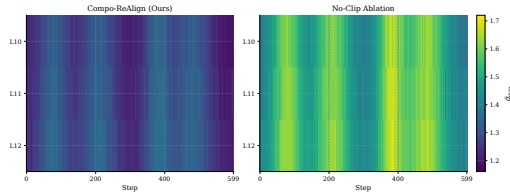 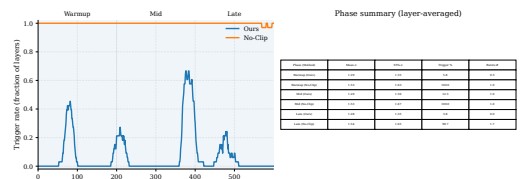

(a) Time–heatmaps of $\widehat{\sigma}_{\max}$ (L10–L12 $\times$ 600 steps). Left: **Compo-ReAlign**; Middle: *No-Clip* ablation; shared colorbar on the right.

(b) Per-step trigger rate (smoothed) with phase bands (left) and an embedded phase summary table (right).

Figure 12: **Training dynamics: sensitivity bursts & clipping responses.** Trust-region clipping suppresses and shortens late-layer spikes, yielding fewer and shorter episodes above $\gamma$.

In the time–heatmaps (Fig. 12a), COMPO-REALIGN exhibits sparse, short spikes confined to early steps and L12, while *No-Clip* shows broad, persistent bands above $\gamma$, especially late. The trigger-rate plot with phase bands (Fig. 12b) indicates a rapid decay and low variance for **Ours**, versus sustained high triggering for *No-Clip*. Together, these visuals show the trust region intercepts sensitivity bursts at critical stages, preventing late-layer geometry blow-ups and stabilizing training.

# D THEORETICAL ANALYSIS

## D.1 IDENTIFIABILITY AND A CRR LOWER BOUND

We formalize when the proposed reversible composer preserves sufficient information to recover the primitive set of a composition and how this yields a lower bound for a structural Compositional Retention Ratio (CRR).

Let $\mathcal{P} = \{u_1, \ldots, u_M\} \subset \mathbb{S}^{d-1}$ denote the *adapted primitive dictionary* with unit vectors $u_j = \frac{\phi(Ae_{p,j})}{\|\phi(Ae_{p,j})\|_2}$. For a composition $S \subset [M]$ of size $m$, define the (unrotated) mean

$$\bar{u}(S) := \frac{1}{m} \sum_{i \in S} u_i, \qquad c(S) := \frac{\bar{u}(S)}{\|\bar{u}(S)\|_2} \in \mathbb{S}^{d-1}. \tag{11}$$

The learned composer applies an orthogonal $R \in O(d)$ (Cayley core) to produce $\widehat{e}_c = R\, c(S)$; since $R$ is known and $R^\top = R^{-1}$, the *canonicalized* composed embedding is $R^\top \widehat{e}_c = c(S)$. We decode primitives by *top-$m$ correlation*:

$$\widehat{S}(c) := \text{Top-}m \text{ indices of } \{\langle c, u_j \rangle\}_{j=1}^M. \tag{12}$$

Define *coherence* $\mu := \max_{i \neq j} |\langle u_i, u_j \rangle| \in [0, 1)$ and the structural CRR for a single composition as

$$\text{CRR}(S) := \frac{|S \cap \widehat{S}(c)|}{m} \in [0, 1]. \tag{13}$$

For brevity write $\bar{u} := \bar{u}(S)$ and $c := c(S)$. We write $a \lesssim b$ to hide absolute constants.

We first show that coherence alone guarantees separation between members and non-members.

**Lemma 1** (Norm of the mean and in/out correlations). *Let $S \subset [M]$ with $|S| = m$ and coherence $\mu$. Then*

$$\|\bar{u}\|_2^2 \geq \frac{1}{m}\Big(1 - (m-1)\mu\Big), \tag{14}$$

*and for any $a \in S, b \notin S$,*

$$\langle \bar{u}, u_a \rangle \geq \frac{1 - (m-1)\mu}{m}, \qquad \langle \bar{u}, u_b \rangle \leq \mu. \tag{15}$$

*Consequently,*

$$\underbrace{\langle c, u_a \rangle}_{\text{member score}} \geq \sqrt{\frac{1 - (m-1)\mu}{m}}, \qquad \underbrace{\langle c, u_b \rangle}_{\text{non-member score}} \leq \mu\sqrt{\frac{m}{1 - (m-1)\mu}}. \tag{16}$$

*Proof.* Since $\|u_i\|_2 = 1$ and $|\langle u_i, u_j \rangle| \leq \mu$ for $i \neq j$,

$$\|\bar{u}\|_2^2 = \frac{1}{m^2} \sum_{i,j \in S} \langle u_i, u_j \rangle \geq \frac{1}{m^2}\Big(m - m(m-1)\mu\Big) = \frac{1}{m}\Big(1 - (m-1)\mu\Big), \tag{17}$$

which is Eq. 14. For any $a \in S$,

$$\langle \bar{u}, u_a \rangle = \frac{1}{m}\Big(\langle u_a, u_a \rangle + \sum_{i \in S \setminus \{a\}} \langle u_i, u_a \rangle\Big) \geq \frac{1}{m}\Big(1 - (m-1)\mu\Big), \tag{18}$$

and for any $b \notin S$,

$$\langle \bar{u}, u_b \rangle = \frac{1}{m} \sum_{i \in S} \langle u_i, u_b \rangle \leq \frac{1}{m} \cdot m\mu = \mu. \tag{19}$$

Divide both by $\|\bar{u}\|_2$ and use Eq. 14 to obtain Eq. 16. $\qquad\square$

**Theorem 1** (Exact identifiability). *Under coherence $\mu < \frac{1}{2m-1}$, for any $S$ with $|S| = m$ the decoding rule satisfies $\widehat{S}(c) = S$. Moreover, the* margin *separating members from non-members obeys*

$$\Delta_0 := \min_{a \in S}\langle c, u_a \rangle - \max_{b \notin S}\langle c, u_b \rangle \geq \frac{1 - (2m-1)\mu}{\sqrt{m}\,\sqrt{1 - (m-1)\mu}}. \tag{20}$$

*The condition $\mu < \frac{1}{2m-1}$ is necessary (up to equality) for uniform separation across all $S$.*

*Proof.* By Lemma 1, for any $a \in S, b \notin S$,

$$\langle c, u_a \rangle - \langle c, u_b \rangle \geq \sqrt{\frac{1 - (m-1)\mu}{m}} - \mu\sqrt{\frac{m}{1 - (m-1)\mu}} = \frac{1 - (2m-1)\mu}{\sqrt{m}\sqrt{1 - (m-1)\mu}}, \tag{21}$$

which is Eq. 20. The right-hand side is positive iff $1 > (2m-1)\mu$, i.e., $\mu < \frac{1}{2m-1}$, which guarantees all members outrank all non-members and hence $\widehat{S}(c) = S$. For necessity, if $\mu \geq \frac{1}{2m-1}$ one can construct $u_i$ with pairwise inner products saturating $\mu$ on two $(m+1)$-tuples such that the bound in Eq. 20 is non-positive, preventing uniform separation for the worst-case $S$. $\qquad\square$

**Remark.** Inequality Eq. 20 attains equality on equiangular tight frames where off-diagonal inner products are constant $\pm\mu$, so the bound is tight in the worst case.

We next allow perturbations in the composed vector before normalization (e.g., training noise or small modeling mismatch). Let the canonical (unrotated) pre-normalized vector be $\bar{u}$ and suppose the model produces

$$\widetilde{c} := \frac{\bar{u} + n}{\|\bar{u} + n\|_2}, \qquad n \in \mathbb{R}^d, \tag{22}$$

so the decoder uses $\widetilde{c}$ in place of $c$.

**Lemma 2** (Lipschitzness of normalization). *If $\|n\|_2 \leq \varepsilon \|\bar{u}\|_2$ with $\varepsilon \in (0,1)$, then*

$$\left\|\widetilde{c} - c\right\|_2 \leq \frac{2\varepsilon}{1-\varepsilon}. \tag{23}$$

*Consequently, for any unit $v \in \mathbb{S}^{d-1}$,*

$$\left|\langle \widetilde{c}, v \rangle - \langle c, v \rangle\right| \leq \frac{2\varepsilon}{1-\varepsilon}. \tag{24}$$

*Proof.* Write $a := \bar{u}$, $x := a + n$, $s := \|a\|_2$, $t := \|x\|_2$. Then

$$\left\|\frac{a}{s} - \frac{x}{t}\right\|_2 \leq \left\|a\left(\frac{1}{s} - \frac{1}{t}\right)\right\|_2 + \left\|\frac{n}{t}\right\|_2 = \frac{|t-s|}{t} + \frac{\|n\|_2}{t} \leq \frac{\|n\|_2}{t} + \frac{\|n\|_2}{t} = \frac{2\|n\|_2}{t}. \tag{25}$$

Since $t \geq s - \|n\|_2 \geq (1-\varepsilon)s$, we obtain $\left\|\widetilde{c} - c\right\|_2 \leq \frac{2\|n\|_2}{(1-\varepsilon)s} = \frac{2\varepsilon}{1-\varepsilon}$. The inner-product bound follows by Cauchy–Schwarz. $\square$

**Theorem 2** (Robust identifiability & deterministic CRR). *Let $\mu < \frac{1}{2m-1}$ and define the clean margin $\Delta_0$ in Eq. 20. If $\|n\|_2 \leq \varepsilon \|\bar{u}\|_2$ with*

$$\varepsilon < \frac{\Delta_0}{4 + \Delta_0}, \tag{26}$$

*then $\widehat{S}(\widetilde{c}) = S$ and hence $\mathrm{CRR}(S) = 1$. More generally, the perturbed margin satisfies*

$$\min_{a \in S} \langle \widetilde{c}, u_a \rangle - \max_{b \notin S} \langle \widetilde{c}, u_b \rangle \geq \Delta_0 - \frac{4\varepsilon}{1-\varepsilon}. \tag{27}$$

*Proof.* By Lemma 2, for any $j$,

$$\left|\langle \widetilde{c}, u_j \rangle - \langle c, u_j \rangle\right| \leq \frac{2\varepsilon}{1-\varepsilon}. \tag{28}$$

Therefore, for any $a \in S$, $b \notin S$,

$$\langle \widetilde{c}, u_a \rangle - \langle \widetilde{c}, u_b \rangle \geq \left(\langle c, u_a \rangle - \langle c, u_b \rangle\right) - \frac{4\varepsilon}{1-\varepsilon} \geq \Delta_0 - \frac{4\varepsilon}{1-\varepsilon}, \tag{29}$$

which is Eq. 27. If the right-hand side is positive then every member still outranks every non-member, so $\widehat{S}(\widetilde{c}) = S$. Solving $\Delta_0 - \frac{4\varepsilon}{1-\varepsilon} > 0$ for $\varepsilon$ yields Eq. 26. $\square$

**A probabilistic CRR lower bound.** To translate perturbations into a CRR bound, suppose $n$ is an isotropic sub-Gaussian vector with parameter $\sigma^2$ (i.e., $\langle n, v \rangle$ is $\sigma$-sub-Gaussian for all $\|v\|_2 = 1$). Using standard norm tails, for some absolute $c > 0$,

$$\Pr\left(\|n\|_2 \geq t\right) \leq 2 \exp\left(-ct^2/\sigma^2\right) \qquad \forall t > 0. \tag{30}$$

Define the *separation radius* $r^\star := \frac{\Delta_0}{4+\Delta_0}\|\bar{u}\|_2$. By Theorem 2, the *pairwise* ranking $\langle \widetilde{c}, u_a \rangle > \langle \widetilde{c}, u_b \rangle$ holds for all $(a,b) \in S \times ([M] \setminus S)$ whenever $\|n\|_2 < r^\star$. Hence, by a union bound over $m(M-m)$ pairs,

$$\Pr\left(\widehat{S}(\widetilde{c}) \neq S\right) \leq m(M-m)\Pr\left(\|n\|_2 \geq r^\star\right) \leq 2m(M-m)\exp\left(-cr^{\star 2}/\sigma^2\right). \tag{31}$$

Using Eq. 14 and Eq. 20,

$$\|\bar{u}\|_2^2 \geq \frac{1 - (m-1)\mu}{m}, \qquad \Delta_0 \geq \frac{1 - (2m-1)\mu}{\sqrt{m}\sqrt{1 - (m-1)\mu}}. \tag{32}$$

Therefore,

$$\boxed{\mathbb{E}\left[\mathrm{CRR}(S)\right] \geq 1 - 2m(M-m)\exp\left(-\frac{c}{\sigma^2} \cdot \frac{\left(\frac{\Delta_0}{4+\Delta_0}\right)^2 \left(1 - (m-1)\mu\right)}{m}\right).} \tag{33}$$

In particular, if $\mu < \frac{1}{2m-1}$ and $\sigma^2 \lesssim \frac{1}{m}\left(1 - (m-1)\mu\right)$, then the failure probability decays exponentially in the dimensionless constant $\left(\frac{\Delta_0}{4+\Delta_0}\right)^2$ and CRR is near 1.

**Dimension–coherence corollary.** If $u_1, \ldots, u_M$ are i.i.d. uniform on $\mathbb{S}^{d-1}$ (or sub-Gaussian normalized), then with probability at least $1 - M^{-2}$,

$$\mu \leq C\sqrt{\frac{\log M}{d}} \tag{34}$$

for an absolute constant $C > 0$. Thus, whenever

$$d \gtrsim (2m-1)^2 \log M, \tag{35}$$

we have $\mu < \frac{1}{2m-1}$ with high probability, and Theorems 1–2 apply. Substituting this $\mu$ into Eq. 20 and Eq. 33 yields explicit $d$–$M$–$m$ trade-offs: the margin scales as $\Delta_0 \gtrsim \frac{1}{\sqrt{m}} - C'(2m-1)\sqrt{\frac{\log M}{md}}$, and CRR concentrates near 1 provided $\sigma^2 \lesssim \frac{1}{m}$.

The orthogonal composer $R$ renders reversibility algorithmic ($R^\top$), while mean aggregation plus low coherence produce a *tight* member/non-member margin Eq. 20. The normalization is stable (Lemma 2), so small perturbations preserve identifiability (Theorem 2). This yields the exponential CRR lower bound Eq. 33, explaining why text-centric buffers that *reduce effective coherence* (semantic diversity) or shrink perturbations (spectral clipping) improve compositional retention.

## E   LLM USAGE

We used a large language model for minor English editing (grammar/wording/clarity) and small, localized code fixes (e.g., resolving syntax errors, adding missing imports). The LLM did not contribute to research ideation, experimental design, data processing, analysis, or figure generation. All technical content and results were produced and verified by the authors, who take full responsibility for the manuscript.

