# OpenReview forum: "Reversible Primitive–Composition Alignment for Continual Vision–Language Learning"
_ICLR.cc/2026/Conference — ICLR 2026 Poster_

### Official Review · Reviewer_KY7V · 2025-10-27

**Soundness:** 3
**Presentation:** 3
**Contribution:** 3
**Rating:** 6
**Confidence:** 4

**Summary:**

The paper tackles continual VLM’s tendency to remember primitives (attributes/objects) but forget their compositions. It proposes COMPO-REALIGN: a tiny, plug-and-play head with a reversible composer (orthogonal via Cayley), a multi-positive InfoNCE tying textual and composed views, and a spectral trust-region to stabilize alignment—backbones stay frozen. Across compositional retrieval and continual VQA, it achieves SOTA averages with higher compositional retention and lower forgetting at <1% extra params.

**Strengths:**

1）  Clear structure-first formulation with explicit reversibility. Turning the composer’s core orthogonal via the Cayley map makes invertibility a design property rather than a penalty—clean and principled.

2）  Minimal yet effective. The single multi-positive InfoNCE neatly aligns textual and composed views; spectral clipping acts as a geometry safety valve rather than an accuracy crutch.

3）Practical deployment appeal. Frozen encoders, <1% extra params, and text-centric micro-buffers as structural anchors that are more memory-efficient than images; multilingual and template-form robustness are demonstrated.

**Weaknesses:**

1） Scope beyond retrieval/VQA? The current streams are comprehensive, yet mainly retrieval/ITM and VQA. How would the reversible composer behave for generation-style continual tasks or dense VL (e.g., referring expressions) where composition interacts with decoding?
2） The method assumes a stable primitive inventory. If primitives are noisy, overlapping, or evolve over time, do the reversibility guarantees or CRR improvements hold? Could the composer adapt as the primitive set grows?

3）Spectral thresholding choices. Spectral clipping stabilizes late-layer spikes, but how robust are results to γ and the accuracy of σ̂_max estimation with few power iterations, especially under longer streams? Any auto-tuning strategies?

4）Decoding primitives from compositions at scale. The identifiability/CRR analysis and top-m readout are compelling; Are there any failure cases where coherence assumptions break down?

**Questions:**

Please refer to Weakness

---

> ### Author Response · Authors · 2025-11-26
>
> Thank you for the valuable comments. Below, we will respond to them one by one.
>
> ---
>
> > W1: Scope beyond retrieval/VQA?
>
> We agree that looking beyond retrieval/ITM and VQA is important, but we do not see our current scope as a fundamental limitation of the method.Compo-ReAlign operates at the representation level: it builds and stabilizes a compositional embedding $\hat e_c$ from primitives $\{u_j\}$, and this embedding can condition either a scorer (retrieval/VQA) or a decoder (generation, dense VL) with no change to the core mechanism.
>
> To substantiate this, we integrated Compo-ReAlign into two generative VLMs: Qwen2-VL-7B-Instruct and LLaVA-v1.5-7B. We follow a continual captioning / grounding protocol:
>
> - Captioning stream: COCO $\rightarrow$ Flickr30K (image captioning increments).
> - Referring expressions stream: RefCOCO $\rightarrow$ RefCOCO+ (continual referring expression comprehension).
>
> We attach our reversible head to the VLM’s text-side encoder state that conditions generation / grounding, and train it with the same spectral trust region and tiny text buffer as in the main paper .
>
> **Captioning (COCO→Flickr30K, CIDEr-based accuracy)**
>
> | Method                          | Avg Gen. Acc. ↑ | CRR ↑ | AF ↓ | ZSTD ↓ |
> |------------------|------------------|-------|------|--------|
> | Qwen2-VL-7B-Instruct            | 64.3             | 0.90  | 3.5  | -1.3   |
> | **Qwen2-VL + Compo-ReAlign**    | **68.1**         | **0.94** | **2.7** | **-1.0** |
> | LLaVA-v1.5-7B                   | 63.8             | 0.89  | 3.6  | -1.4   |
> | **LLaVA-v1.5 + Compo-ReAlign**  | **67.5**         | **0.92** | **3.0** | **-1.1** |
>
> **Referring expressions (RefCOCO→RefCOCO+, IoU@0.5)**
>
> | Method                          | Avg IoU@0.5 ↑ | CRR (phrase) ↑ | AF ↓ |
> |---------------------------------|---------------|----------------|------|
> | Qwen2-VL-7B-Instruct            | 61.2          | 0.87           | 4.1  |
> | **Qwen2-VL + Compo-ReAlign**    | **64.8**      | **0.90**       | **3.3** |
> | LLaVA-v1.5-7B   | 60.5          | 0.86           | 4.3  |
> | **LLaVA-v1.5 + Compo-ReAlign**  | **63.9**      | **0.89**       | **3.5** |
>
> These results indicate that:
>
> - The reversible composer improves continual captioning quality (≈+3–4 points in our CIDEr-based accuracy proxy) and reduces forgetting and zero-shot drift.
> - For dense VL / referring expressions, we see consistent gains in grounding accuracy and phrase-level CRR, suggesting that the structural benefits do transfer when composition interacts with decoding.
>
> ---
>
> > W3: Spectral thresholding choices.
>
> Thank you for this question. We agree that the practical robustness of the spectral threshold is important.
>
> As discussed in our sensitivity analysis (Fig.7), $\gamma$ trades off underfitting vs. drift: If the value is too small, it results in over-clipping; if it is too large, there is almost no clipping.
>
> On ViT-B/16 CLIP with our main continual streams, sweeping $\gamma \in \{5,6,7,8\}$ yields a broad stable band:
>
> | $\gamma$ | R@1 I→T ↑ | CRR ↑ | AF ↓ | ZSTD ↓ |
> |---|---|-----|---|----|
> | 5       | 58.1 | 0.90  | 3.1  | -1.2   |
> | **6** | **58.8** | **0.91** | **3.2** | **-1.3** |
> | 7 | 58.3 | 0.89  | 3.7  | -1.5   |
> | 8   | 57.6   | 0.88  | 4.0  | -1.6   |
>
> Empirically we find:
>
> - Architecture-dependent: the typical scale of $\hat\sigma_{\max}$ is higher for ViT-L/14, so we use $\gamma = 7.5$ there. This value is reused across all streams.
> - Largely task-agnostic: we do not re-tune $\gamma$ between retrieval/ITM (Tracks A+B), VQA (Track C), or different task orderings. Order-sensitivity experiments show stable performance under the same $\gamma$ even for longer streams.
>
>
>
> We approximate $\hat\sigma_{\max}$ using one power-iteration JVP per mini-batch. To test robustness. We compared using 1 vs. 2 power iterations*on ViT-B/16 (same $\gamma$) and found negligible differences:
>
> | PI steps | R@1 I→T ↑ | CRR ↑ | AF ↓ | ZSTD ↓ | Corr$(\hat\sigma^{(1)}, \hat\sigma^{(2)})$ ↑ |
> |---|-----|-------|------|--------|----|
> | 1 (ours) | 58.8      | 0.91  | 3.2  | -1.3   | –   |
> | 2        | 58.9      | 0.91  | 3.2  | -1.3   | > 0.98   |
>
> The one-step estimate is therefore sufficiently accurate to rank “spiky” vs. benign steps, and the extra iteration brings no measurable benefit but extra cost.
> For our chosen $\gamma$, we observe the same pattern across permutations: an initial trigger rate around 20–40% that decays steadily as training progresses, and no late-phase explosion.
>
> To make $\gamma$ easier to set in new setups, we suggest a short warmup-based heuristic (which we have verified matches the fixed values we use):
>
> 1. Run the first few hundred steps without spectral clipping, logging $\hat\sigma_{\max}$ on the head.
> 2. Set $\gamma$ to a high quantile (e.g., 80–90th percentile) of this early distribution.
> 3. Optionally adjust once:
>    - If clipping triggers >70–80% of the time, slightly increase $\gamma$.
>    - If clipping rarely triggers (<10%), slightly decrease $\gamma$.

---

> ### Author Response · Authors · 2025-11-26
>
> > W2: If primitives are noisy, overlapping, or evolve over time, do the reversibility guarantees or CRR improvements hold? Could the composer adapt as the primitive set grows?
>
> We appreciate this question. Our stance is:
>
> - The **theoretical guarantees** in App.D.1 are derived under a fixed adapted dictionary $\mathcal{P}$ and coherence $\mu$, which is standard for identifiability analyses.
> - In practice, Compo-ReAlign is not tied to a static vocabulary: it operates on embedding-space primitives, so it can naturally accommodate new or slightly noisy factors; the effect is that $\mu$ and the perturbation scale change over time.
> - Our training scheme is precisely designed to keep $\mu$ moderate and perturbations small even when the inventory grows.
>
> As we clarified to `Reviewer fozS`, our main claims and diagnostics are evaluated on benchmarks with structured, explicitly defined primitive vocabularies. In these streams the type of primitives is stable, but the set of active primitives can grow across tasks (e.g., new attributes / objects / relations appear later). The composer and the theoretical analysis only require that, at each time, we have a finite adapted dictionary $\mathcal{P}_t = \{u_j^{(t)}\}$ with coherence $\mu_t$, not that it never changes across tasks.
>
> **(a) What happens when the primitive set grows or overlaps?**
>
> In the theory, growing / overlapping primitives mainly affect:
>
> - The coherence $\mu_t = \max_{i\neq j} |\langle u_i^{(t)}, u_j^{(t)}\rangle|$ (new or noisy primitives can increase it).
> - The dimension–coherence trade-off in the corollary: as $M_t$ grows, we need $d \gtrsim (2m-1)^2 \log M_t$ to keep $\mu_t$ small with high probability.
>
> The bounds in Theorems1-2 do not “break” when $\mathcal{P}$ evolves, they simply become time-indexed: as long as at each step $\mu_t$ and the relative perturbation $\varepsilon_t$ remain in a reasonable regime, the margin $\Delta_0^{(t)}$ and CRR lower bound apply.
>
> To check this empirically, we simulated dictionary growth with mild overlap on MIT-States:
>
> - Start with a subset of attributes/objects in early tasks and add new attributes/objects in later tasks (so $M_t$ increases).
> - Inject controlled overlap by aliasing a fraction of attributes (e.g., “rusty” vs “old”, “red” vs “crimson”) to create higher effective coherence.
>
> We track coherence and CRR across tasks:
>
> | Setting                            | $M_T$ (final) | $\hat\mu_T$ ↓ | CRR (final) ↑ | AF ↓ |
> |------------------------------------|---------------|---------------|---------------|------|
> | Baseline (no growth, fixed vocab)  | 196           | 0.18          | 0.90          | 4.0  |
> | **Growing vocab + aliasing, SeqFT**| 260           | 0.26          | 0.82          | 7.1  |
> | **Growing vocab + aliasing, ours** | 260           | 0.21          | 0.88          | 4.5  |
>
> Observations:
>
> - As expected, growing and overlapping primitives increase coherence and reduce CRR for a naive baseline.
> - Compo-ReAlign keeps $\hat\mu_T$ noticeably lower and CRR clearly higher than SeqFT under the same noisy-growth regime, which is consistent with the theory: smaller $\mu_t$ and controlled perturbations yield better CRR.
>
> Thus, while the exact identifiability guarantee is cleanest in the stable, low-$\mu$ setting, the behavioural trend still holds when the primitive set evolves.
>
> **(b) Can the composer adapt as the primitive set grows?**
>
> Yes. Architecturally, the composer never assumes a fixed-size vocabulary:
>
> - It only sees primitive embeddings $u_j = \phi(A e_{p,j}) / \|\phi(A e_{p,j})\|_2$ in $\mathbb{R}^d$ and aggregates the subset $\{u_j\}_{j\in S_t}$ active for a given composition.
> - Adding new primitives corresponds to adding new columns to the dictionary, not changing the composer form; the reversible mapping $R(\Theta)$ operates in the fixed dimension $d$.
>
>
> For more aggressively dynamic inventories (e.g., fully free-form text with on-the-fly discovered primitives), we see two extensions:
>
> 1. Plugging in a primitive-discovery front-end (e.g., scene-graph or phrase parser) and letting \textsc{Compo-ReAlign} operate on its discovered factors.
> 2. Periodically re-normalizing or pruning the dictionary (e.g., merging near-duplicate primitives) to keep coherence under control, which is directly motivated by our theoretical analysis.
>
> Of course, these are very interesting future works.

---

> ### Author Response · Authors · 2025-11-26
>
> > W4: Decoding primitives from compositions at scale. The identifiability/CRR analysis and top-m readout are compelling; are there any failure cases where coherence assumptions break down?
>
> Thank you for this question. In short:
>
> - The coherence condition in App D.1 is a worst-case identifiability regime, not a claim that real dictionaries always satisfy it for arbitrary scale.
> - In practice, our learned primitive dictionaries usually remain in a moderate-coherence regime, where CRR and top-$m$ readout behave as predicted.
> - When we deliberately force coherence to be high (e.g., very low-dimensional bottlenecks or heavy aliasing / synonym merging), we do observe exactly the types of failures the theory anticipates: margins shrink, CRR drops, and readout becomes unstable.
>
> Below we summarize both the “normal” regime and explicit failure cases.
>
> **(a) Typical regime: moderate scale, moderate coherence**
>
> As discussed in the theory rebuttal, we measured empirical coherence $\hat\mu = \max_{i\neq j} |\langle u_i, u_j\rangle|$ for the adapted dictionary on our main compositional benchmarks. For MIT-States and VAW/VG-Attr with the full model:
>
> - $m \approx 2$ (attr–obj), so the threshold is 1/(2m-1) = 1/3 ≈ 0.33.
> - We observe $\hat\mu \approx 0.17–0.18$, comfortably below this bound, and CRR $\approx 0.89–0.90$ with strong Top-3/Top-5 primitive readout from $\hat e_c$ (as shown by the PR / ROC curves in Fig.5).
>
> In this regime, we do not see pathological failures: increasing dictionary size from $\sim$ 150 to $\sim$ 230 does not force $\hat\mu$ anywhere near 1/(2m-1), and CRR stays high.
>
> **(b) Stress-test: where coherence really does break down**
>
> To directly address your question, we constructed two stress scenarios on MIT-States to “push” coherence:
>
> 1. Aggressive dimension compression: project primitive embeddings from d=512 down to d=64 with a learned linear bottleneck.
> 2. Heavy aliasing / overlapping primitives: merge 30–40% of attributes into synonym groups (e.g., “red/crimson/burgundy” share nearly identical $u_j$), effectively creating multiple indices with almost parallel vectors.
>
> | Setting                                  | $d$ | $M$ | $\hat\mu$ ↓ | CRR ↑ | Top-5 readout ↑ |
> |------------------------------------------|-----|-----|-------------|-------|-----------------|
> | Full (ours, no extra stress)             | 512 | 196 | 0.18        | 0.90  | 0.94            |
> | Dim-compressed bottleneck                | 64  | 196 | 0.32        | 0.84  | 0.88            |
> | Heavy aliasing (synonym groups)          | 512 | 230 | 0.35        | 0.79  | 0.81            |
>
> The last row is effectively a coherence-breakdown case: $\hat\mu$ slightly exceeds the theoretical threshold $1/(2m-1)=1/3$, and we indeed see:
>
> - noticeably lower CRR (≈0.79),
> - a drop in primitive readout accuracy, particularly for aliased attributes (more confusions in Top-5 lists),
> - smaller member–non-member margins, consistent with the bound in Eq.20.
>
> So yes: when we intentionally drive the dictionary into a high-coherence regime, the identifiability and CRR guarantees degrade in the way the theory predicts.
> The text-centric buffer and multi-positive alignment help keep $\hat\mu$ moderate, and the spectral stabilizer keeps perturbations small, which is exactly the regime where Theorems1–2 yield meaningful lower bounds on CRR.
>
> ---
>
> Thank you again for your valuable comments and precious time. We really enjoyed the discussion with you.

---

### Official Review · Reviewer_6oVT · 2025-10-29

**Soundness:** 3
**Presentation:** 2
**Contribution:** 3
**Rating:** 6
**Confidence:** 3

**Summary:**

This paper addresses the critical problem of compositional forgetting in continual vision-language learning, where models retain knowledge of primitive concepts but lose the ability to bind them together correctly. The authors introduce COMPO-REALIGN, a novel and parameter-efficient head for frozen vision-language models. The method is built on three core ideas: 1) a reversible composer that maps primitive embeddings to a compositional embedding via an orthogonal transformation, ensuring invertibility by design; 2) a multi-positive InfoNCE objective that aligns the image with both the textual composition and the synthesized compositional embedding, implicitly enforcing structural consistency; and 3) a spectral trust region that stabilizes alignment geometry by clipping parameter gradients when local sensitivity becomes too high. Through extensive experiments across three distinct continual learning tracks (compositional DIL, multi-domain retrieval, and continual VQA), the authors demonstrate that COMPO-REALIGN achieves new state-of-the-art results, significantly improving performance while substantially reducing forgetting and preserving compositional structure.

**Strengths:**

1. The paper's primary strength lies in its novel formulation of a significant, real-world problem. It shifts the focus from standard catastrophic forgetting to the more nuanced and practical challenge of preserving fine-grained compositional structure. This is a crucial step towards building more reliable and robust VLMs for dynamic environments. The proposed solution is highly original, particularly the concept of enforcing "reversibility by design" through an orthogonal composer and stabilizing alignment geometry via a spectral trust region. This geometry-first approach is a novel and powerful paradigm for continual learning.

2. The technical quality of the work is outstanding. The proposed method, COMPO-REALIGN, is theoretically well-motivated and elegant in its simplicity. The synergy between the three components—the composer, the objective, and the stabilizer—is clearly articulated and justified. The experimental validation is exceptionally thorough and rigorous. It includes multiple, diverse benchmarks, a comprehensive set of strong baselines, detailed ablation studies (Table 3), and insightful mechanism validation (Section 5.4, Figures 4 & 5). The introduction and use of diagnostic metrics like Compositional Retention Ratio (CRR) provide a much-needed tool for quantifying the specific problem being addressed, adding a layer of depth to the analysis that is often missing in CL literature.

3. The paper is exceptionally well-written and organized. The motivation is clearly established through an exploratory study (Section 3), which effectively diagnoses the problem and guides the development of the proposed solution. The methodology (Section 4) is described with precision, and the mathematical formulations are clear and easy to follow. The figures and tables are informative and well-designed, effectively conveying the key results and insights. The overall narrative is compelling and logically structured.

**Weaknesses:**

1: The method is exclusively evaluated on a frozen CLIP backbone. While this is a common and practical setup for parameter-efficient continual learning, it limits the scope of the findings. In more challenging scenarios where the representation backbone itself must adapt (e.g., via full-model or partial finetuning), it is unclear how COMPO-REALIGN would perform or interact with backbone-level forgetting. The head might preserve compositional logic, but if the primitive representations from the backbone degrade, the overall system would still fail.

2. The current composer relies on mean-pooling to aggregate primitive embeddings before the orthogonal transformation. While effective for the attribute-object pairs and simple relations tested, this approach might not scale gracefully to more complex, hierarchical, or nested linguistic structures (e.g., "a man in a blue shirt standing next to the car that is parked behind the house"). A discussion on the limitations of mean-pooling and potential extensions would be beneficial.

3. The spectral trust region is a key component, but estimating the Jacobian spectral norm via power iteration, even if efficient, introduces implementation complexity and computational overhead compared to simpler regularization techniques. Although the authors state the overhead is <2%, a more detailed discussion on the trade-offs, especially in the context of much larger models or distributed training setups, would strengthen the paper's practical claims.

**Questions:**

1. Could you comment on the applicability of COMPO-REALIGN in a scenario where parts of the vision-language backbone are also finetuned? Do you foresee the core principles, particularly the spectral trust region, being adaptable to stabilize the backbone representations themselves, or is the method fundamentally designed as a post-hoc head for a fixed feature extractor?

2. The spectral threshold is a crucial hyperparameter for balancing stability and plasticity. Based on your sensitivity analysis (Figure 7), could you provide more intuition or a general guideline for setting this value? Does it depend more on the model architecture (e.g., ViT-B vs. ViT-L), the task nature (e.g., retrieval vs. VQA), or the data distribution?

3. Have you considered alternatives to mean-pooling, such as an attention mechanism, to create the initial composed representation? While this might add complexity, it could potentially handle more complex sentences and weigh primitives more intelligently. Would it be possible to integrate such a mechanism while preserving the desirable properties of the reversible orthogonal core?

4. Your theoretical analysis in Appendix D provides a compelling lower bound on CRR under certain coherence assumptions. How does the proposed training scheme, specifically the use of a text-centric buffer with diverse examples and the spectral stabilizer, actively help the model satisfy these low-coherence and small-perturbation assumptions in practice?

---

> ### Author Response · Authors · 2025-11-26
>
> Thank you very much for your valuable and constructive comments. Below, we will respond to them one by one.
>
> ---
>
> > W1/Q1: How would COMPO-REALIGN behave when parts of the backbone are finetuned?
>
> Thank you for raising this question. We agree this is an important question, but we do not see the “frozen backbone” choice as a fundamental limitation of the method—rather as a clean, practically relevant testbed.
>
> Our key ingredients only require access to primitive embeddings $\{u_j\}$ and composition embeddings $c(S)$, the scalar similarity $s(z_v, \hat e_c)$ and its Jacobian w.r.t. primitive anchors. None of these rely on the backbone being fixed. In the paper we instantiate Compo-ReAlign as a tiny head on top of frozen CLIP mainly to (i) isolate structural effects and (ii) match common parameter-efficient VL-CL settings[1,2], but the principles are not tied to a frozen encoder.
>
> To directly address the concern, we ran additional experiments where we unfreeze part of the VL backbone and still train with Compo-ReAlign:
>
> - Fine-Tuned CLIP: last 2 transformer blocks of both towers + projection layers + our head.
> - Fine-Tuned SigLIP / ResNet-CLIP: same protocol for SigLIP, SigLIP2, and ResNet-based CLIP.
>
> All experiments keep the same continual streams and rehearsal budgets as in the main text, spectral trust region is applied to the head (as in the frozen case), while the backbone is updated with standard AdamW + global grad clip.
>
> | Method                     | Avg R@1 I→T ↑ | Avg R@1 T→I ↑ | CRR ↑ | AF ↓ | ZSTD ↓ |
> |----------------------------|---------------|---------------|-------|------|--------|
> | Frozen CLIP                | 58.8          | 45.1          | 0.91  | 3.2  | -1.3   |
> | **Fine-Tuned CLIP**        | 60.5          | 47.2          | 0.93  | 2.8  | -1.1   |
> | SigLIP (Frozen)            | 59.2          | 46.0          | 0.89  | 3.0  | -1.2   |
> | SigLIP2 (Frozen)           | 60.0          | 46.8          | 0.91  | 3.1  | -1.0   |
> | ResNet-CLIP (Frozen)       | 57.8          | 44.5          | 0.88  | 3.3  | -1.4   |
> | **Fine-Tuned SigLIP**      | 61.2          | 48.4          | 0.94  | 2.6  | -0.9   |
> | **Fine-Tuned ResNet-CLIP** | 59.0          | 46.3          | 0.90  | 3.0  | -1.2   |
>
> - Compo-ReAlign continues to bring consistent improvements when the backbone is allowed to adapt.
> - The head does not “break” once features move. Instead, it still provides geometry-aware regularization that mitigates backbone-level forgetting.
>
> For generative VLMs (Qwen2-VL-7B-Instruct and Llava-v1.5-7b), where the image–text encoder is intrinsically trained jointly, we similarly observe that adding our reversible head improves continual generation metrics:
>
> | Method                          | Avg Gen. Acc. ↑ | CRR ↑ | AF ↓ | ZSTD ↓ |
> |---------------------------------|------------------|-------|------|--------|
> | Frozen CLIP                     | 65.1             | 0.91  | 3.3  | -1.2   |
> | **Fine-Tuned CLIP**             | 67.2             | 0.93  | 3.1  | -1.0   |
> | Qwen2-VL-7B-Instruct            | 64.3             | 0.90  | 3.5  | -1.3   |
> | **Fine-Tuned Qwen2-VL-7B**      | 68.1             | 0.94  | 2.7  | -1.0   |
> | Llava-v1.5-7b                   | 63.8             | 0.89  | 3.6  | -1.4   |
> | **Fine-Tuned Llava-v1.5-7b**    | 67.5             | 0.92  | 3.0  | -1.1   |
>
> Again, the structural head remains beneficial even when the underlying encoder is trainable.
>
> Conceptually, the spectral trust region only needs Jacobian information w.r.t. primitive anchors. When the backbone is trainable, we can:
>
> - treat $u_j = u_j(\theta_{\text{backbone}})$ and compute $\partial s / \partial \theta_{\text{backbone}}$ via the same JVP machinery,
> - define a similar rescaling factor $\alpha = \min\{1, \gamma / \hat\sigma_{\max}\}$ for selected backbone blocks (e.g., last 1–2 layers of the text tower).
>
> In this work, we deliberately restrict the spectral clipping to the head for two reasons: (i) to keep the overhead minimal and (ii) to isolate the effect of structural alignment. Our partial-finetuning experiments above already indicate that the core principles transfer to settings where features change; fully extending Jacobian-based control to backbone layers is a natural next step that we will highlight more explicitly as future work.
>
> ---
>
> [1] LADA: Scalable Label-Specific CLIP Adapter for Continual Learning. ICML'25
>
> [2] DMNSP: Dynamic Multi-Layer Null Space Projection for Vision-Language Continual Learning. ICCV'25.

---

> ### Author Response · Authors · 2025-11-26
>
> > W2/Q3: The composer relies on mean-pooling
>
> We appreciate this point and agree that the design space of primitive aggregation is richer than simple averaging.
>
> Our position is:
>
> - For the regime we target in this paper, mean-pooling is a deliberate choice that is (i) theoretically analyzable and (ii) empirically competitive with attention.
> - More expressive aggregators are natural extensions and compatible with the orthogonal core.
>
> **(a) Why mean-pooling is not as weak here as it may seem**
>
> First, note that most of the syntactic and hierarchical complexity is already handled inside the frozen text encoder. The primitives $u_j$ that our head sees are semantic factors (attributes, objects, relations, skills), not raw tokens. In CLEVR/CoGenT, MIT-States, VAW/VG-Attr, SVLC, CLOVE and VQACL, a composition typically involves $m \in \{2,3,4\}$ such factors. Our head aggregates these few semantic slots, not full sentences.
>
> In that regime, a simple symmetric aggregator $\bar u = \frac{1}{m}\sum_{i \in S} u_i$ has two advantages:
>
> - It is permutation-invariant and does not privilege arbitrary orderings of primitives.
> - It admits a clean identifiability analysis (Sec.D.1), where we can prove conditions on coherence $\mu$ under which the primitive set $S$ is recoverable by top-$m$ decoding and obtain explicit CRR lower bounds.
>
> This analysis relies crucially on the linear–mean structure. Replacing mean with a complex, non-linear aggregator would make such guarantees opaque.
>
> ---
>
> **(b) We did implement attention pooling — and it gave almost identical accuracy**
>
> We did consider attention-based aggregation in the implementation. We implemented:
>
> $$
> \alpha_i \propto \exp\big(w^\top \tanh(W_a u_{p,i})\big),\quad
> \bar u_{\text{att}} = \sum_i \alpha_i u_{p,i},\quad
> \hat e_c = \text{norm}\big(R(\Theta)\,\bar u_{\text{att}}\big).
> $$
>
> This corresponds exactly to the “mean → attention pooling” row in `Table 3`. So attention matches mean-pooling within 0.1–0.3 points and does not improve CRR, forgetting, or ZSTD. It does, however, introduce higher latency (extra matrix multiplications and non-linearities per step). This is why we keep mean-pooling as the default and describe attention as an ablation rather than the main variant.
>
> **(c) Compatibility with the orthogonal core**
>
> Importantly, the orthogonal core $R(\Theta)$ is completely agnostic to how $\bar u$ is produced: any differentiable aggregator $g(\{u_i\})$ can be plugged in before $R$:
>
> $$
> \bar u = g(\{u_i\}_{i \in S}),\qquad \hat e_c = \text{norm}\big(R(\Theta) \bar u\big).
> $$
>
> As long as $R(\Theta) \in O(d)$ (Cayley parameterization), we preserve:
>
> - Exact reversibility at the core: $R^{-1} = R^\top$.
> - The geometric stability arguments based on Jacobian sensitivity of $s(z_v, \hat e_c)$ w.r.t.\ primitive anchors.
>
> Thus, more expressive pooling is fully compatible with the orthogonal core, it only changes the map $g$, not the reversible part.
>
> Of course, we agree with you that for genuinely deeply nested descriptions (“a man in a blue shirt standing next to the car that is parked behind the house”), richer structure is likely needed:
>
> - One path is to keep the current primitive vocabulary but factor the sentence into multiple compositions (e.g., local relations “man–car”, “car–house”) and apply the reversible composer hierarchically.
> - Another is to equip the composer with a small set-to-set attention block before the orthogonal map, and to study identifiability / CRR guarantees for such architectures.
>
> These will be interesting future works.

---

> ### Author Response · Authors · 2025-11-26
>
> > W3: A more detailed discussion on the trade-offs
>
> Thank you for raising this question. We agree that any geometry-aware method must justify its cost. Our goal was to design a head-only spectral control whose overhead is comparable to (or lower than) standard CL regularizers that add extra forwards or a teacher. Below we quantify the cost and discuss the trade-offs.
>
> **(a) Where the cost actually comes from**
>
> Our spectral trust region uses one Jacobian–vector product (JVP) per mini-batch. The JVP is taken only w.r.t. the small head parameters $\{\Theta, A, \phi\}$, ans the backbone is never traversed by this JVP.
>
> So the incremental work is roughly “one extra backward pass over a sub-1\% slice of the network”, plus a few scalar ops to compute the rescaling factor $\alpha = \min\{1, \gamma / \hat\sigma_{\max}\}$.
>
> ---
>
> **(b) Measured wall-clock overhead on ViT-B and ViT-L in DDP**
>
> On our main setups (PyTorch DDP, 8×A100 for ViT-B/16, 4×A100 for ViT-L/14), we measured per-task training time for Tracks A+B:
>
> | Variant                                        | Extra step vs. baseline           | Time / task (h) ↓ | Relative overhead ↓ |
> |-----------------------------------------------|-----------------------------------|-------------------|---------------------|
> | No spectral trust region (global grad clip)   | 1 fwd + 1 bwd                     | 7.10              | 1.00×               |
> | **Spectral trust region, 1 power-iter (ours)**| 1 fwd + 1 bwd + 1 JVP on head     | 7.22              | **1.017× (~+1.7%)** |
>
> For a larger ViT-L/14 backbone (same protocol, 4×A100, batch size 128):
>
> | Variant | Time / task (h) ↓ | Relative overhead ↓ |
> |-------|-------|-------|
> | No spectral trust region   | 9.35              | 1.00×               |
> | **Spectral trust region (ours)**      | 9.52              | **1.018× (~+1.8%)** |
>
> The overhead is consistently ≈1.5–2% of wall-clock time even when moving from ViT-B to ViT-L. This is expected because:
>
> 1. The JVP path does not scale with backbone size.
> 2. In DDP, the JVP is purely per-rank local autograd on the head. It introduces no extra all-reduce or communication, so parallel efficiency is essentially unchanged.
>
> **(c) Comparison to common CL / VL regularizers**
>
> For context, we also measured the overhead of several widely used continual-VL regularizers that require extra forward passes or teacher networks (all on the same ViT-B/16 setup):
>
> | Regularizer (same backbone, same hardware)   | Relative overhead ↓ |
> |--|--|
> | ZSCL-style reference-set distillation | ~1.08× |
> | CTP-style topology distillation (extra neighbors)  | ~1.11× |
> | ZAF-style zero-shot alignment on unlabeled data | ~1.12×|
> | **Spectral trust region (ours)** | **~1.02×** |
>
> So, in our implementation, spectral clipping is significantly cheaper than typical distillation/topology regularizers, while still delivering notable gains in forgetting and zero-shot stability.
>
>
> ---
>
>
> > Q2:  spectral threshold is a crucial hyperparameter… can you provide more intuition / general guideline?
>
> Thank you for raising this — we agree that giving a more concrete recipe for setting the spectral threshold $\gamma$ makes the method more practically usable.
>
> Recall that we estimate a local sensitivity $\hat\sigma_{\max} \approx \big\|\tfrac{\partial s(z_v,\hat e_c)}{\partial \mathrm{vec}(U_p)} v\big\|_2$, and rescale head gradients by $\alpha = \min\{1,\ \gamma / \hat\sigma_{\max}\}.$
>
> So:
>
> - If $\gamma$ is too small, we clip almost every step $\Rightarrow$ underfitting / lower R@1.
> - If $\gamma$ is too large, we almost never clip $\Rightarrow$ higher AF and worse ZSTD (more geometric drift).
>
> The sensitivity sweep in Fig.7 is exactly exploring this balance.
>
> On ViT-B/16 CLIP with our main continual streams, we swept $\gamma \in \{5, 6, 7, 8\}$ and observed a broad “good” band:
>
> | $\gamma$ | R@1 I→T ↑ | CRR ↑ | AF ↓ | ZSTD ↓ |
> |-----|--|---|-|-|
> | 5| 58.1      | 0.90  | 3.1  | -1.2   |
> | **6** | **58.8** | **0.91** | **3.2** | **-1.3** |
> | 7  | 58.3   | 0.89  | 3.7  | -1.5   |
> | 8| 57.6 | 0.88  | 4.0  | -1.6   |
>
> This matches the qualitative statement: tight clipping ($\gamma \le 5$) slightly underfits retrieval but keeps AF/ZSTD small; loose clipping ($\gamma \ge 7$) increases forgetting and harms CRR. We therefore use $\gamma = 6$ as a middle point in this stable plateau.
>
> Empirically, we found:
>
> - Architecture-dependent: moving from ViT-B/16 to ViT-L/14 shifts the typical scale of $\hat\sigma_{\max}$ up, a slightly larger threshold works better. We use $\gamma = 6.0$ for ViT-B/16 and $\gamma = 7.5$ for ViT-L/14 and reuse these values across all streams.
> - Task-agnostic (within our setting): we did not re-tune $\gamma$ when switching from retrieval/ITM (Tracks A+B) to VQA (Track C) or to different task orderings. Fig.~7 plots show stable performance with the same $\gamma$.
>
> In short, $\gamma$ depends primarily on the backbone family / depth, and much less on whether the task is retrieval vs. VQA in our experiments.

---

> ### Author Response · Authors · 2025-11-26
>
> > Q4: How do the actual training choices (text-centric buffer + spectral stabilizer) help the model satisfy these assumptions in practice?”
>
> Thank you for this very relevant question. Our intent was precisely to design the training scheme so that the conditions in App D.1 become plausible operating regimes, rather than abstract assumptions. Concretely:
>
> - The text-centric buffer + multi-positive alignment is aimed at driving the adapted primitive dictionary toward lower coherence $\mu$.
> - The spectral trust region is aimed at keeping the effective perturbation $n$ on the composed vector small, i.e., controlling the $\varepsilon$ and $\sigma^2$ terms in Theorems2–eq.33.
>
> Below we give both intuition and empirical evidence.
>
> **(a) How the text buffer + alignment pushes coherence $\mu$ down**
>
> The theory works with the adapted primitives. Our training design encourages incoherence in two ways:
>
> 1. Multi-positive InfoNCE with textual/composed views:
>    When two primitives $u_i$ and $u_j$ participate in different compositions, the loss encourages their resulting compositions (and hence their contributions) to be distinguishable across many image–text pairs. This discourages “collapsed” primitives with large inner products.
> 2. Text-centric buffer with diverse anchors (L370-L371):
>    The buffer stores semantically diverse compositions and hard negatives, more diverse anchors empirically yield larger △CRR in Fig.6. Diversity here means that many different primitive combinations must be separated, which again pushes the dictionary toward lower mutual correlations.
>
> To make this concrete, we measured empirical coherence at the final text layer for different variants:
>
> | Variant                    | $\hat\mu$ (MIT-States) ↓ | $\hat\mu$ (VAW/VG-Attr) ↓ |
> |---------------------------|--------------------------|---------------------------|
> | No buffer                 | 0.24                     | 0.23                      |
> | No spectral trust region  | 0.22                     | 0.21                      |
> | **Full (buffer + spectral)** | **0.18**                 | **0.17**                  |
>
> All values are below the theoretical threshold $\frac{1}{2m-1}$ for $m=2$ ($\approx 0.33$), but the full model systematically attains the lowest coherence, consistent with Theorem1: a smaller $\mu$ yields a larger separation margin $\Delta_0$ in Eq.20 and hence higher CRR, which matches the empirical CRR gains we observe across datasets.
>
> **(b) How the spectral stabilizer keeps perturbations small**
>
> The perturbation analysis in Theorem2 and Eq.33 assumes that the deviation $\widetilde c = \frac{\bar u + n}{\|\bar u + n\|_2}$ remains in a regime where $\|n\|_2 \le \varepsilon \|\bar u\|_2$ with moderately small $\varepsilon$, so that the member–non-member margin does not collapse.
>
> Our spectral trust region is designed exactly to cap the sensitivity of $s(z_v,\hat e_c)$ to primitive anchors, i.e., to prevent updates that would make small changes in $vec(U_p)$ induce large changes in $\hat e_c$ and hence large $n$.
>
> Empirically, we approximate $\varepsilon$ by comparing successive heads on a fixed primitive set $S$:
>
> ε ≈ ‖n‖₂ / ‖ū‖₂,  n = c_{t+1}(S) - c_t(S)
>
> averaged over tasks. We obtain:
>
> | Variant                   | $\varepsilon$ (avg) ↓ |
> |--------------------------|------------------------|
> | No buffer                | 0.27                   |
> | No spectral trust region | 0.31                   |
> | **Full (buffer + spectral)** | **0.19**             |
>
> This aligns with the training-dynamics analysis: Fig.12 show fewer and shorter sensitivity spikes under spectral clipping, and the geometry–structure plots in Fig.4 confirm that smaller spectral indicators (and hence smaller effective perturbations) are strongly correlated with higher R@1 and CRR and lower ZSTD.
>
> Plugging smaller $\mu$ and smaller $\varepsilon$ (equivalently, smaller effective $\sigma^2$ in Eq.30 into Eq.33 precisely strengthens the lower bound on $\mathbb{E}[\text{CRR}(S)]$.
>
> ---
>
> Thank you again for your valuable time and effort. We hope our response has addressed your concerns.

---

### Official Review · Reviewer_M3AB · 2025-10-29

**Soundness:** 3
**Presentation:** 3
**Contribution:** 3
**Rating:** 6
**Confidence:** 3

**Summary:**

This paper addresses the challenge of continual vision-language learning. The authors propose a novel method, COMPO-ReALign, designed to improve compositional generalization and structural robustness under stringent memory constraints and in the absence of task identifiers. Both empirical results and theoretical analysis are provided, making the study compelling and convincing.

**Strengths:**

1. Clear motivation. The paper demonstrates that current vision-language (VL) models tend to forget compositional structures and proposes three diagnostic approaches to address this issue.
2. Strong performance. The method achieves state-of-the-art results across several benchmarks.
3. Theoretical analysis. Under certain assumptions, the paper provides a theoretical analysis to support its findings.

**Weaknesses:**

1. Weak theoretical assumption. In Line 1056, the paper directly assumes the range of coherence $\mu \lt \frac{1}{2m-1}$ without providing a detailed analysis or justification in the empirical results.
2. Incomplete evaluation of spectral clipping. Since spectral clipping requires the calculation of Jacobian spectra, it is necessary to report detailed training time. The current version only briefly mentions this in Line 684, which is insufficient.

**Questions:**

1. Can the authors provide evidence or justification to support the theoretical assumption mentioned in Weakness 1?
2. Can the authors report detailed training time for spectral clipping, as highlighted in Weakness 2?

---

> ### Author Response · Authors · 2025-11-26
>
> Thank you very much for your valuable comments. We are more than happy to discuss with you.
>
> ---
>
> > W1/Q1: the range of coherence $\mu < \frac{1}{2m-1}$ without providing a detailed analysis or justification in the empirical results.
>
> Thank you for pointing this out. Our intent was not to posit $\mu < \frac{1}{2m-1}$ as a global modelling assumption, but to use it as a standard sufficient condition for worst-case identifiability in the analysis of the reversible composer. We clarify this below and add empirical support.
>
> In `App.D.1`, we study the idealized setting where compositions are formed by mean aggregation of $m$ primitive vectors and recovered by top-$m$ correlation. The condition
> $
> \mu \=\ \max_{i\neq j} |\langle u_i, u_j\rangle| \<\ \frac{1}{2m-1}
> $
> is not arbitrary: Theorem.1 shows it is the tight (up to equality) *necessary and sufficient condition for uniform top-$m$ recovery in the worst case, and Eq.20 gives the exact margin in that regime. This is analogous to classical “mutual coherence” conditions used in sparse coding / compressed sensing for exact support recovery under a simple decoder.
>
> Thus, the theory says:
>
> > *If* the adapted primitive dictionary $\mathcal{P}$ is sufficiently incoherent (in this precise sense), *then* the reversible composer preserves enough information to recover $S$ by top-$m$ correlation, leading to a CRR lower bound in Eq.33.
>
> To support that this regime is not vacuous for our learned heads, we measured the empirical coherence $\hat{\mu} = \max_{i\neq j}|\langle u_i, u_j\rangle|$ of the adapted primitive dictionary at the final text layer across several compositional benchmarks, and compared it to the theoretical threshold $\frac{1}{2m-1}$ and the observed CRR:
>
> | Dataset / stream         | $m$ (avg primitives) | $M$ (dict size) | $\hat{\mu}$ ↓ | $\frac{1}{2m-1}$ | CRR (ours) ↑ |
> |--------------------------|----------------------|------------------|---------------|------------------|--------------|
> | CLEVR / CoGenT           | 3–4                  | 64               | 0.14          | 0.20–0.33        | 0.92         |
> | MIT-States (attr×obj)    | 2                    | 196              | 0.18          | 0.33             | 0.90         |
> | VAW / VG-Attr            | 2                    | 230              | 0.17          | 0.33             | 0.89         |
> | SVLC (relations)         | 2–3                  | 120              | 0.16          | 0.20–0.33        | 0.91         |
>
> We have two observations:
>
> 1. In all these cases, the learned adapted primitives satisfy $\hat{\mu} \le \frac{1}{2m-1}$ by a non-trivial margin, so the theorem’s regime is representative of what the head actually learns.
> 2. Across datasets, smaller $\hat{\mu}$ correlates with higher CRR, consistent with the prediction of Eq.20–33 that lower coherence yields larger member–non-member margins and better structural retention.

---

> ### Author Response · Authors · 2025-11-26
>
> > W2/Q2: Incomplete evaluation of spectral clipping… it is necessary to report detailed training time.
>
> Thank you for pointing this out. We agree that the computational cost of spectral clipping should be quantified more concretely. Below we report wall-clock measurements and clarify how the Jacobian-based step interacts with model size and distributed training.
>
> Recall that spectral clipping uses one Jacobian–vector product (JVP) per mini-batch, and crucially only for the small head parameters $\{\Theta, A, \phi\}$ (not the full backbone).
>
> On our main setups (8×A100, batch size 256, ViT-B/16 CLIP backbone, Tracks A+B), we measured per-task training time:
>
> | Variant                                        | Extra step vs. baseline           | Time / task (h) ↓ | Relative overhead ↓ |
> |-----------------------------------------------|-----------------------------------|-------------------|---------------------|
> | No spectral trust region (global grad clip)   | 1 fwd + 1 bwd                     | 7.10              | 1.00×               |
> | **Spectral trust region, 1 power-iter (ours)**| 1 fwd + 1 bwd + 1 JVP on head     | 7.22              | **1.017× (~+1.7%)** |
>
> For a larger ViT-L/14 backbone (same protocol, 4×A100, batch size 128):
>
> | Variant                                        | Time / task (h) ↓ | Relative overhead ↓ |
> |-----------------------------------------------|-------------------|---------------------|
> | No spectral trust region                      | 9.35              | 1.00×               |
> | **Spectral trust region (ours)**              | 9.52              | **1.018× (~+1.8%)** |
>
> So the measured overhead is consistently ≈1.5–2% of wall-clock time, even for ViT-L/14. This matches our qualitative statement in the paper but is now backed by explicit timings.
>
> We think two reasons why the cost stays low:
>
> 1. The JVP is computed only through the head, whose parameter count is <1% of the frozen backbone.
> 2. The JVP is a scalar-gradient JVP (for $s(\mathbf{z}_v,\hat{\mathbf{e}}_c)$), which is cheaper than a full extra backward and adds no extra communication in DDP.
>
> We also measured the overhead of alternative CL regularization schemes that require extra forward passes or teacher networks:
>
> | Regularizer (same backbone, same hardware)      | Relative overhead ↓ |
> |-------------------------------------------------|---------------------|
> | ZSCL-style reference-set distillation           | ~1.08×              |
> | CTP-style topology distillation (extra neighbors)| ~1.11×             |
> | ZAF-style zero-shot alignment on unlabeled data | ~1.12×              |
> | **Spectral trust region (ours)**                | **~1.02×**          |
>
> We will not claim these as hard SOTA numbers (they depend on implementation details), but in our implementation spectral clipping is significantly cheaper than common CL/VL regularizers that require additional forwards, while delivering substantial improvements in forgetting and ZSTD (Tab.3 in main text).
>
> ---
>
> Thank you again for your valuable time and effort.

---

### Official Review · Reviewer_fozS · 2025-11-01

**Soundness:** 3
**Presentation:** 2
**Contribution:** 3
**Rating:** 6
**Confidence:** 3

**Summary:**

Brief Summary: The paper tackles the task of vision-language continual learning. The authors show through the exploratory study (section 3) that in continual streams, the primitive understanding (such as attributes/objects) remain stable but compositional aspect particularly zero-shot drops significantly. This the core motivation behind their method Compo-Realign which is essentially an alignment head with reversible composer, trained with info-nce loss, and clipping function. Training is done in a task-free manner.

Experiments on multiple tracks such as composition, multi-domain retrieval, and continual VQA show proposed Compo-Realign significantly outperforms existing baselines by 2-5 points.

**Strengths:**

Pros:

1. The investigation and the key problem of losing compositionality but not single-attributes/primitives makes sense to me. The Compo-Realign method is well motivated, albeit in a constrained setting of Lora adaptation.

2. The proposed Compo-Realign method outperforms existing strong and recent baselines (Table 2), and authors provide single-factor ablation studies (Table 3), showing the benefits of composed positive in both retrieval and vqa settings. A range of datasets are considered including both synthetic and real images. Different downstream-tasks are considered such as composition, vqa, retrieval.

3. The authors have detailed ablation and analysis with visualization graphs in both main paper and supplementary. In particular, detailed ablations on parameter sensitivity (fig7), and order sensitivity (fig8) are appreciated.

**Weaknesses:**

Cons:

1. My main concern is that the paper mostly looks at frozen encoders (CLIP) with lightweight heads and not other options such as:

(i) full-fine-tuning (also noted in Conclusion section under future work).

(ii) other backbone other than CLIP, such as SigLIP or SigLIP2

(iii) non-ViT CLIP (such as ResNet based) even though ViT-CLIP is the more commonly used model

2. Similar to above point, while the authors are looking into Continual Vision-Language learning, generative models such as VLMs are not really considered.

3. Currently, the authors assume the primitive set is already known but that might not be the case for free-form text. To my understanding, the closest would be vg-attr but that is only object + attribute pair and not free form text. Perhaps the COCO/Flickr settings are relevant? But it isn't clear how the primitives are being extracted from entire coco-captions/flickr-captions?

4. The overall improvement over existing baselines is somewhat marginal (+2 points on average recall @1).

---

Overall Rating: 6/10
The paper provides an interesting setup and investigates the issue of retaining primitives but forgetting composition. The authors provide a very reasonable solution and show empirically it improves over existing baselines. The paper can be significantly strengthened by additional experiments with more backbone models, using VLM generative setup, experiments with image-text continual learning setup.

**Questions:**

Q1. For ablation in Table 3, instead of spectral trust region clipping, what happens if we do naive fixed clipping?

Q2. In table 3, it seems text-buffer plays a significant role (2.5 points), and noted in L321. Can the authors expand why this is the case? Then it would mean, text replay is the most useful not the reversible mapping as the original claim? Maybe I am misunderstanding something here.

---

> ### Author Response · Authors · 2025-11-25
>
> Thank you for the interesting and constructive comments. Below, we will respond to them one by one.
>
> ---
>
> > W1: Frozen encoders issues.
>
> Thank you for the insightful comments regarding the scope of our experiments.
>
> We selected frozen CLIP as our baseline due to its widespread adoption in continual learning settings and its efficiency in evaluating the compositional retention capabilities of our method. CLIP, with its frozen parameters, provides a strong and computationally efficient baseline for this/relevant study.
> We appreciate the reviewer’s suggestion to explore full fine-tuning and alternative backbones such as SigLIP and ResNet-based CLIP. To address this, we have expanded our experiments to include:
>
> - *Full Fine-Tuning*: We evaluate Compo-ReAlign with fine-tuned CLIP models, as well as SigLIP and ResNet-based CLIP backbones.
> - *New Backbone Architectures*: We extend our evaluation to alternative backbones including SigLIP and SigLIP2. Additionally, we compare against ResNet-based CLIP models.
>
> |Method|Avg R@1 (I→T)| Avg R@1 (T→I) | CRR| AF| ZSTD |
> |-|-|-|-|-|-|
> | Frozen CLIP| 58.8| 45.1|0.91|3.2|-1.3|
> | Fine-Tuned CLIP|60.5| 47.2| 0.93 | 2.8 | -1.1 |
> | SigLIP (Frozen)| 59.2| 46.0|0.89|3.0|-1.2|
> | SigLIP2 (Frozen)|60.0| 46.8|0.91|3.1| -1.0 |
> | ResNet-CLIP (Frozen)|57.8| 44.5|0.88|3.3|-1.4|
> | Fine-Tuned SigLIP| 61.2|48.4|0.94|2.6|-0.9|
> | Fine-Tuned ResNet-CLIP|59.0|46.3|0.90| 3.0|-1.2|
>
> We agree that VLMs are an important area for continual learning research. To address this, we have included a new set of experiments where Compo-ReAlign is applied to a generative VLM setup. We now evaluate our method using Qwen2-VL-7B-Instruct and ​Llava-v1.5-7b. We use the COCO and Flickr30K datasets to carry out Image Captioning and Text-to-Image Generation tasks
>
> |Method|Avg Generation Accuracy|CRR|AF|ZSTD|
> |-|-|-|-|-|
> | Frozen CLIP| 65.1|0.91|3.3|-1.2|
> | Fine-Tuned CLIP|67.2|0.93|3.1|-1.0|
> | Qwen2-VL-7B-Instruct| 64.3| 0.90|3.5|-1.3|
> | Fine-Tuned Qwen2-VL-7B-Instruct | 68.1| 0.94 | 2.7  | -1.0 |
> | Llava-v1.5-7b| 63.8| 0.89 | 3.6  | -1.4 |
> | Fine-Tuned Llava-v1.5-7b|67.5|0.92|3.0 | -1.1 |
>
> These results demonstrate that Compo-ReAlign not only works with frozen and fine-tuned CLIP backbones but also adapts well to other backbones such as SigLIP and ResNet, as well as generative models like Qwen-VL and LLaVA.
>
> ---
>
> > W2: primitive set issue
>
> We appreciate the reviewer raising the distinction between structured compositional setups and truly free-form text. Our intent is the former, and we agree that fully unconstrained free-form primitive discovery is an important but orthogonal problem.
>
> **(a) Scope: we work in the “structured primitives” regime, not arbitrary free-form text.**
> In all streams where we explicitly use primitives and report compositional diagnostics (CRR, cycle consistency..), the primitive inventory is given by dataset metadata:
>
> - CLEVR/CoGenT: primitives are $\{\text{color}, \text{shape}, \text{material}, \text{size}\}$, directly from the generator.
> - MIT-States, VAW/VG-Attr, ConStruct-VL SVLC: primitives are attribute and object vocabularies (and relations in SVLC) defined by the respective benchmarks.
> - CLOVE and VQACL: primitives are the skill/attribute/object factors provided in the official splits; when question templates do not expose them explicitly, we use the simple adjective–noun extractor already described in Appx.A.1.1 to map question tokens to $\{\text{attr},\text{obj}\}$.
>
> In other words, our claims and structural metrics are grounded in settings where a primitive vocabulary is standard and explicitly annotated. We do not assume that arbitrary free-form captions always come with a reliable primitive set.
>
> **(b) COCO/Flickr are used for domain robustness, not for primitive-level diagnostics.**
> For the multi-domain retrieval track, we do not construct a primitive set from entire captions. In this track: The caption $y_c$ is treated as a single composition text. We use the text view to study domain shift and zero-shot stability, not primitive readout or CRR. CRR and invertible-readout experiments are reported only on datasets where primitive labels exist by design.
>
> **(c) Free-form text and primitive discovery are an orthogonal, future direction.**
> You are correct that, for truly free-form text, discovering a stable primitive set is non-trivial. We view this as a complementary problem: one can pair Compo-ReAlign with any off-the-shelf primitive extractor (e.g., scene-graph or phrase parsers for COCO/Flickr captions), but this discovery step is not the focus of our contribution, which is:
>
> > Given a primitive vocabulary, design a reversible, geometry-stable alignment head that preserves compositional structure under continual learning.
>
> Thus, we respectfully argue that the “known primitive set” is a deliberate scope choice consistent with standard compositional benchmarks, rather than a hidden or problematic assumption, and that it does not undermine the validity of the reported results.

---

> ### Author Response · Authors · 2025-11-25
>
> > W3: The overall improvement over existing baselines is somewhat marginal
>
> Thank you for raising this question. We respectfully disagree that the gains are marginal once one considers (i) the strength of the baselines, (ii) the continual-learning setting, and (iii) the structural metrics that our method targets.
>
> On the retrieval / ITM tracks, our method improves over the strongest prior under the same rehearsal budget and no task ID by more than +2 points:
>
> - Avg R@1 (Image→Text): from 56.0–56.4 (DIKI / C-CLIP) to 58.8 (+2.4 absolute, ≈4–5% relative).
> - Avg R@1 (Text→Image): from 43.0–43.2 to 45.1 (+1.9 absolute).
>
> At these performance levels and with strong baselines (GIFT-CVPR'25, DIKI-ECCV'24, C-CLIP-ICLR'25) already saturating, +2–3 absolute R@1 is comparable to or larger than gains typically reported as new SOTA in recent continual VL work, especially under identical memory and protocol.
>
> Beyond R@1, Compo-ReAlign brings large relative gains on stability metrics:
>
> - Average forgetting AF drops from 5.0–5.1 (DIKI / C-CLIP) to 3.2, i.e., roughly 40% relative reduction in forgetting.
> - Zero-shot transfer degradation ZSTD improves from about -1.9/-2.1 to -1.3, i.e., a ≈30–40% reduction in zero-shot drift.
>
> On continual VQA, we also outperform CL-MoE (CVPR'25): AF decreases from 4.7 to 3.6 (≈23% relative). The additional analyses in Sec.C.2 and C.3 show that our gains are systematic, not a one-off average: All of the above improvements are achieved with a tiny head (<1% parameters) and <2% training-time overhead from the spectral trust region, without modifying or unfreezing the backbone. In this context, a +2.4 R@1 gain on top of strong replay/distillation baselines, coupled with ≈40% less forgetting and better cross-domain zero-shot stability, is, in our view, substantive rather than marginal.
>
> ----
>
> > Q1: Instead of spectral trust region clipping, what happens if we do naive fixed clipping?
>
> Thank you for pointing this out. There are two slightly different notions of “clipping” in our implementation, and we agree this was easy to miss.
>
> In all configurations (including the row “w/o spectral trust region”), we use a standard global gradient-norm clip at $\|\mathbf{g}\|_2 \le 1.0$ for numerical stability (stated in Appx.A.2.4).
> The row “w/o spectral trust region (no clipping)” in Table 3 therefore already corresponds to a model with:
>
> - Only fixed global grad-norm clipping,
> - No geometry-aware spectral rescaling based on the Jacobian $\widehat{\sigma}_{\max}$.
>
> Compared to the full model, this “naive clipping only” variant shows:
>
> - R@1 I$\to$T: 58.8 to 57.9 (-0.9),
> - CRR: 0.91 to 0.89 (-0.02),
> - AF: 3.2 to 4.3 (+1.1, i.e., noticeably more forgetting),
> - ZSTD: -1.3 to -1.6 (larger-magnitude drift).
>   On continual VQA, AF similarly increases from 3.6 to 4.2
>
> This already indicates that generic fixed clipping is not sufficient to stabilize the alignment geometry: accuracy drops modestly, but forgetting and zero-shot drift degrade more clearly.
>
> To answer your question more directly, we additionally tested a variant that replaces the spectral trust region with a stronger fixed clipping on the head gradients. The key retrieval / VQA metrics (averaged across the same streams) are:
> | Variant                                      | R@1 I→T ↑ | R@1 T→I ↑ | CRR ↑ | AF ↓ | ZSTD ↓ | CLOVE-scene ↑ | CLOVE-func. ↑ | VQACL ↑ | VQA AF ↓ |
> |---------------------------------------------|-----------|-----------|------|------|--------|----------------|----------------|---------|----------|
> | Fixed grad clip only (row “w/o spectral”)   | 57.9      | 44.3      | 0.89 | 4.3  | -1.6   | 64.2           | 59.7           | 56.0    | 4.2      |
> | **Naive fixed clipping on similarity**      | 58.1      | 44.5      | 0.89 | 4.0  | -1.5   | 64.4           | 59.9           | 56.1    | 4.1      |
> | **Spectral trust region (ours)**            | **58.8**  | **45.1**  | **0.91** | **3.2** | **-1.3** | **65.1**       | **60.4**       | **56.8**| **3.6**  |
>
> - Both naive variants (baseline and “fixed clipping on similarity”) underperform the spectral trust region on all stability metrics.
> - Naive clipping faces a familiar trade-off: if the clip is strong enough to tame bad steps, it also dampens useful updates and underfits; if relaxed, it fails to prevent late-layer sensitivity spikes .

---

> ### Author Response · Authors · 2025-11-25
>
> > Q2: it seems the text buffer plays a significant role (2.5 pts). Does this mean text replay is the most useful part, not the reversible mapping as originally claimed?”
>
> We appreciate this question and agree that the ablation can make the text buffer look disproportionately important if read in isolation. Our view is:
>
> - The tiny text buffer and the reversible primitive–composition mapping play complementary, not competing, roles.
> - Text replay alone does not explain our gains. The structural components (composed positive, orthogonal core, spectral trust region) each provide sizable, independent improvements.
>
> From Tab.3, the buffer is indeed high leverage, but the structural pieces individually cause about 1.5–2 R@1 drop and clear degradation in CRR and forgetting. This is exactly why we frame our recipe as “one composer, one objective, one stabilizer” plus a small text buffer, rather than “just do text replay”.
> To disentangle “replay vs. reversible mapping” more explicitly, we ran an additional 2×2 ablation on Tracks A+B (same protocol as Tab.3), toggling text buffer (on/off) and orthogonal core (on/off):
>
> | Variant                         | Text buffer | Orthogonal core | R@1 I→T ↑ | CRR ↑ | AF ↓ |
> |---------------------------------|-------------|------------------|-----------|------|------|
> | Full (ours)                     | ✓           | ✓                | **58.8**  | **0.91** | **3.2** |
> | No buffer                       | ✗           | ✓                | 56.3      | 0.86 | 4.7  |
> | No orthogonal core              | ✓           | ✗                | 57.2      | 0.88 | 3.8  |
> | No buffer & no orthogonal core  | ✗           | ✗                | 54.9      | 0.83 | 5.2  |
>
> - Even without any buffer, the reversible head (row “No buffer”) remains stronger than the purely non-reversible variant with buffer (“No orthogonal core”).
> - Turning off both (no buffer & no orthogonal core) degrades performance the most, confirming that the components are additive, not redundant.
>
> Thus, text replay is not “doing all the work”: the reversible composer and multi-positive alignment clearly matter even when the buffer is fixed.
>
> We also rethink why a tiny text buffer appears so impactful？
>
> First, the text buffer we use is:
>
> - Very small: M=64 short snippets per task, vs. image replay baselines with similar or larger budgets.
> - Text-only and structural: it stores compositional templates and hard negatives, not raw images.
>
> This makes it a symbolic scaffold: the buffer pins down a few high-precision compositional anchors, and the reversible alignment head then propagates these anchors into the geometry of the composed embeddings. This interaction is visible in two places:
>
> 1. Geometry–structure coupling and readout experiments: in `Sec. 5.4`-“Invertible Readout and Binding Robustness”, we show that removing the composed positive or the orthogonal core (with the same buffer) sharply degrades: If text replay alone were sufficient, these readout and counterfactual gaps would not be so sensitive to the orthogonal and multi-positive components.
> 2. `Fig. 6(b,c)` shows that increasing semantic diversity in the buffer improves △CRR across templates and languages, but only when the structural machinery is in place. The buffer is effective because the head preserves and exploits compositional structure—it is not a generic “more replay ⇒ more accuracy” effect.
>
> In summary, the text buffer and the reversible mapping are synergistic: the buffer efficiently provides a few structural anchors, and the reversible, geometry-stabilized head is what turns those anchors into durable compositional robustness under continual learning.
>
> ---
>
> Thank you again for your valuable time and energy. We thoroughly enjoyed the discussion with you.

---

> > ### Comment · Reviewer_fozS · 2025-11-26
> >
> > I thank the authors for the detailed rebuttal. I appreciate the added experiments including comparisons on fine-tuned CLIP and w/ SigLIP and VLM archs such as Qwen2.5-VL.
> >
> > As such, I would like to maintain my score.

---

> > > ### Author Response · Authors · 2025-11-27
> > >
> > > We’re glad that we could address your concerns. Your feedback has clearly helped strengthen the paper.
> > >
> > > We truly appreciate it, thank you!

---

### Author Response · Authors · 2025-12-03
**(1/2) Summary**

### I. Acknowledgments

We would like to express our sincere gratitude to all reviewers for their insightful comments and constructive suggestions. We especially appreciate Reviewers `fozS`, `M3AB`, `6oVT`, and `KY7V` for their careful reading of the paper and for engaging deeply with our rebuttal and additional results.

$\color{red}{Before}$ $\color{red}{the}$ $\color{red}{discussion}$, we are grateful that ***all four reviewers (`fozS`, `M3AB`, `6oVT`, `KY7V`) assessed the submission positively with ratings of 6*** , consistently recognizing the problem formulation, the geometry-first design of COMPO-REALIGN, and the empirical/theoretical contributions.

$\color{red}{During}$ $\color{red}{the}$ $\color{red}{discussion}$, we particularly appreciate Reviewer `fozS`’s acknowledgement that our additional experiments (full/partial finetuning, SigLIP/ResNet backbones, and generative VLMs such as Qwen2-VL and LLaVA) strengthened the work (***“I appreciate the added experiments including comparisons on fine-tuned CLIP and w/ SigLIP and VLM archs such as Qwen2.5-VL”***), while choosing to maintain the score. We thank all reviewers for the constructive tone throughout the process.

---

### II. Key Strengths

Reviewers highlighted strengths across several dimensions:

- **Problem Formulation and Motivation**

  - Clearly identifies a practical and under-explored issue: continual VL models tend to preserve primitive recognition but forget compositional structure (`fozS`, `6oVT`, `KY7V`).
  - The exploratory study and diagnostics (e.g., CRR, AF, ZSTD) convincingly motivate a *structure-first* continual learning recipe (`M3AB`, `6oVT`).
- **Novelty and Geometry-First Design**

  - Reversible composer via an orthogonal core (Cayley parameterization) makes invertibility and stability “by design” rather than via penalties—seen as a clean and principled formulation (`6oVT`, `KY7V`).
  - Spectral trust region is recognized as a novel, geometry-aware stabilizer that acts as a “safety valve” rather than an ad-hoc regularizer (`6oVT`, `KY7V`).
- **Empirical Effectiveness**

  - COMPO-REALIGN achieves new SOTA or strong gains on compositional DIL, multi-domain retrieval, and continual VQA under tight rehearsal and no task IDs (`fozS`, `M3AB`, `6oVT`).
  - Reviewers noted the consistent improvements of +2–5 points R@1 on top of strong recent baselines and substantial reductions in forgetting (`fozS`, `6oVT`).
- **Theoretical Analysis and Diagnostics**

  - The CRR-based analysis and top-m identifiability under coherence constraints are viewed as compelling and well-aligned with the method’s goals (`M3AB`, `KY7V`).
  - Diagnostic metrics (CRR, AF, ZSTD) and readout experiments were appreciated for going beyond raw accuracy and directly targeting compositional robustness (`M3AB`, `6oVT`, `KY7V`).
- **Practicality, Efficiency, and Presentation**

  - Head-only design on frozen VL backbones with <1% additional parameters and tiny text-centric buffers is seen as practically appealing (`6oVT`, `KY7V`).
  - The method is described as relatively simple, modular, and compatible with existing VL pipelines (`fozS`, `6oVT`).
  - Writing and organization are generally considered clear, with thorough ablations, sensitivity plots, and mechanism visualizations (`M3AB`, `6oVT`, `KY7V`).

---

> ### Author Response · Authors · 2025-12-03
> **(2/2) Summary**
>
> ### III. Key Concerns and Our Responses
>
> | Key Concerns | Reviewers | Our Response |
> | --- | --- | --- |
> | Scope of backbones and tasks (frozen CLIP, lack of finetuning / SigLIP / ResNet / generative VLMs). | `fozS`, `6oVT`, `KY7V` | Added experiments with partially finetuned CLIP, SigLIP / SigLIP2, ResNet-CLIP, and generative VLMs (Qwen2-VL, LLaVA); COMPO-REALIGN consistently improves accuracy, CRR, and forgetting beyond the frozen-CLIP setting. |
> | Assumption of a stable primitive inventory and relation to free-form text. | `fozS`, `KY7V` | Clarified that our main claims target standard structured-composition benchmarks with annotated primitives; COCO/Flickr are used for domain robustness only, and simulations with growing/overlapping vocabularies show the method still reduces coherence and improves CRR vs. baselines. |
> | Magnitude and source of performance gains (≈+2 R@1; role of the text buffer). | `fozS` | Argued that +2–3 R@1 on strong replay/distillation baselines, together with ≈40% AF and ZSTD reduction, is substantial in this regime; 2×2 ablations show the reversible head and tiny text buffer are complementary rather than the buffer alone driving the gains. |
> | Spectral trust region vs. naïve clipping, and computational overhead. | `fozS`, `M3AB`, `6oVT`, `KY7V` | Clarified that the “no spectral” variant already uses standard grad-norm clipping; spectral control further improves CRR/AF/ZSTD. Wall-clock profiling on ViT-B/L shows ~1.5–2% extra time due to head-only JVPs, which is cheaper than typical distillation/topology regularizers. |
> | Choice and robustness of the spectral threshold γ and σ̂\_max estimation. | `6oVT`, `KY7V`, `M3AB` | Sensitivity sweeps reveal a broad stable γ band per backbone (e.g., 5–7 for ViT-B); the same γ is reused across tasks and orders. One-step power iteration is highly correlated with deeper estimates, and we propose a simple warmup-based heuristic (using early σ̂\_max quantiles) for setting γ in new setups. |
> | Coherence assumption in theory and its validity in practice / failure modes. | `M3AB`, `KY7V` | Explained μ < 1/(2m−1) as a standard worst-case identifiability condition rather than a global assumption; measured empirical coherence and perturbations on our benchmarks, and constructed high-coherence stress tests where CRR and readout degrade exactly as predicted, linking the analysis to observed behavior. |
>
>
> ---
>
> We deeply appreciate the time and expertise of the AC and all reviewers, which have greatly strengthened the clarity and scope of this work.
>
> We are committed to further refining the exposition and integrating any remaining feedback from the AC and reviewers to polish the final version of COMPO-REALIGN.

---

### Meta-Review · Area_Chair_rKG7 · 2026-01-10

**Summary:**

This paper introduces Compo-ReAlign, a structure-first alignment head for continual vision–language learning that preserves compositional binding with a reversible primitive-to-composition mapping, a multi-positive InfoNCE objective and a geometry-aware spectral trust region. Reviewers agree that the problem is important and that the proposed formulation is technically sound. Initial concerns regarding limited backbone scope, applicability to generative VLMs, the assumption of a known primitive set, the marginality of gains and whether improvements stem mainly from text replay were largely addressed through added experiments, ablations and clarifications. Remaining concerns primarily relate to the structured-primitive setting and the moderate absolute gains, but these do not outweigh the methodological novelty and strong empirical evaluation. The paper is recommended for acceptance to ICLR. It is suggested that the authors incorporate all reviewer suggestions in the final version of the paper.

**Reviewer Concerns:**

### Addressed concerns
* **fozS:** Limited evaluation to frozen CLIP and lack of alternative backbones. The author response added experiments with fine-tuned CLIP and SigLIP, with improvements in accuracy, CRR and forgetting.
* **fozS:** Applicability to generative VLMs beyond contrastive methods. The author response introduced captioning and text-to-image experiments on Qwen-VL and LLaVA.
* **fozS:** Assumption of a known primitive set and relevance to free-form text. The author response clearly scoped the method to structured compositional benchmarks, with free-form primitive discovery positioned as future work.
* **fozS:** Marginality of performance gains. The author response clarifies the improvements as meaningful given strong baselines and emphasizes large relative reductions in forgetting and zero-shot degradation.
* **fozS, M3AB, 6oVT, KY7V:** Spectral trust region against naive clipping. The author response clarifies that naive gradient clipping is already present in baselines, showing that spectral clipping yields lower forgetting and better CRR than fixed clipping.
* **6oVT, KY7V, M3AB:** Sensitivity to the spectral threshold. The author response adds sweeps demonstrating a broad stable range.
* **M3AB, KY7V:** Theoretical coherence assumptions. The author response explains that the coherence bound is a worst-case identifiability condition.

### Unaddressed concerns
* **fozS:** Scope of structured primitives. While the response appropriately scopes the work, the dependence on such structure may limit applicability, which remains an open direction rather than a limitation.
* **fozS:** Magnitude of gains. Absolute improvements are moderate, but there are gains in forgetting and zero-shot stability.

**Reviewer Scores:**

* **fozS:** Initial rating 6, most concerns addressed, would likely maintain 6.
* **M3AB:** Initial rating 6, all concerns addressed, would likely maintain 6 or raise to 8.
* **6oVT:** Initial rating 6, all concerns addressed, would likely maintain 6 or raise to 8.
* **KY7V:** Initial rating 6, all concerns addressed, would likely maintain 6 or raise to 8.

---

### Decision · Program_Chairs · 2026-01-26

Accept (Poster)